# ExploraCoder: Advancing code generation for multiple unseen APIs via planning and chained exploration

## Abstract

Through training on publicly available source code libraries, large language models (LLMs) can invoke multiple encapsulated APIs to solve complex programming problems. However, existing models inherently cannot generalize to use APIs that are unseen in their training corpora. As libraries continuously evolve, it becomes impractical to exhaustively retrain LLMs with new API knowledge. This limitation hampers LLMs from solving problems which require newly introduced or privately maintained libraries. Human programmers often explore unfamiliar APIs by writing experimental code before invoking them for a more complex problem. Inspired by this behavior, we propose **ExploraCoder**, a training-free framework that empowers LLMs to invoke multiple unseen APIs in code solution by (1) planning a complex problem into several API invocation subtasks, and (2) exploring correct API usage through a novel chain-of-API-exploration. Concretely, ExploraCoder guides the LLM to iteratively generate several experimental API invocations for each simple subtask, where the promising execution experience are exploited by subsequent subtasks. This forms a chained exploration trace that ultimately guides LLM in generating the final solution. We evaluate ExploraCoder on Torchdata-Github benchmark as well as a newly constructed benchmark that involves more complex API interactions. Experimental results demonstrate that ExploraCoder significantly improves performance for models lacking prior API knowledge, achieving an absolute increase of 11.24% over naive RAG approaches and 14.07% over pretraining methods in pass@10. Moreover, the integration of a self-debug mechanism further boosts ExploraCoder's performance on more challenging tasks. Comprehensive ablation and case studies provide further insights into the effectiveness of ExploraCoder. [1].

## 1 Introduction

Library-oriented code generation refers to the automatic generation of code that leverages library APIs to solve specific programming problems (Zan et al., 2022; Liu et al., 2023). This task poses a challenge in recent AI research due to the complexity of diverse API usage and the interactions between different APIs (Alrubaye et al., 2019; Zan et al., 2024). Recent success of large language model (LLM), such as ChatGPT (OpenAI, 2022) and CodeLlaMA (Rozière et al., 2024), has demonstrated remarkable capability in various code generation tasks (Yan et al., 2024). Researchers have found that LLMs can effectively invoke APIs by pretraining on vast amounts of contemporary public libraries (Zan et al., 2023). However, a persistent challenge arises when the target API knowledge is sparse, outdated, or entirely unseen in the training data. This limitation hampers LLMs from problem solving that requires newly introduced or privately maintained libraries.

Prior work proposed to use continual pretraining (Gururangan et al., 2020) techniques to address this knowledge gap (Zan et al., 2022). But it is often impractical to exhaustively retrain LLMs since the code libraries are continuously evolving. And training data is especially hard to collect for newly introduced libraries. Another line of work adopts a naive retrieval-augmented generation

---

[1]Data and code are available at `https://anonymous.4open.science/r/ExploraCoder_paper`

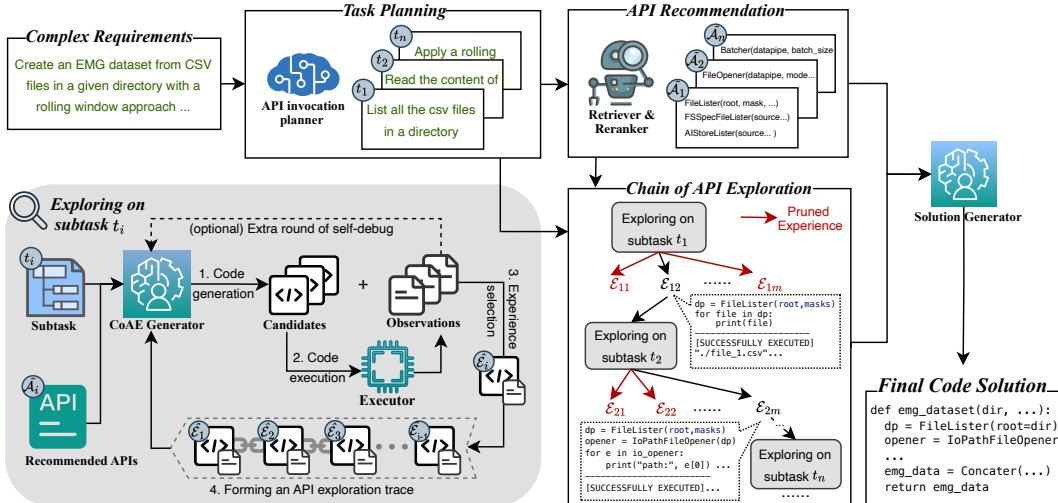

Figure 1: An Overview of ExploraCoder Framework. ExploraCoder processes the given problem through *Task Planning*, *API Recommendation*, and *Chain of API Exploration* modules. The grey block in the bottom-left corner illustrates the detailed exploration process in the Chain of API Exploration. Finally, the processed results are used by a solution generator to generate final code solutions for the programming problem.

(RAG) framework for unseen-library-oriented API invocations (Zan et al., 2023; Zhou et al., 2023; Liu et al., 2023), where the code generation process is divided into two phases: A retriever model retrieves relevant APIs from library documents. Then a generator model generates a code solution based on the problem description and retrieval results. However, they are mainly designed for simple API invocation tasks. Recent studies have shown that these existing approaches exhibit limited performance in complex programming tasks that require multiple API invocations (Zan et al., 2023; 2024; Ma et al., 2024b).

We propose to solve this issue by optimizing the RAG framework, as current solutions fall short on two fronts. On the one hand, it is hard for retrievers to map a comprehensive user requirement to a diverse range of APIs with various functionalities. On the other hand, existing generators lack the reasoning capability to handle intricate interactions among multiple APIs, especially when the models have not been trained on their usage knowledge.

Exploratory programming (Sheil, 1986; Beth Kery & Myers, 2017) is a paradigm where programmers learn to do an unfamiliar task by actively experimenting with different possibilities using code. Specifically, when human programmers are tasked to solve a problem using an unfamiliar library. they would first learn the library's capability from documents, then experiments with relevant API calls to gain hands-on usage experience.

Inspired by this behavior, we propose ExploraCoder, a training-free framework aiming to facilitate LLM to invoke multiple unseen APIs through planning and a chain of API exploration (CoAE). As shown in Figure 1, given a complex programming problem, ExploraCoder first plans a series of simple API invocation subtasks based on library document. And a set of relevant APIs is recommended for each subtask. Then, a chained API exploration is carried out to iteratively experiments with various API invocations and pass on valuable usage experience to subsequent subtasks. This process forms an API exploration trace, which serves as a chain-of-thought (CoT) (Wei et al., 2022) instruction to guide the LLM in generating the final code solution.

We evaluate ExploraCoder on an existing multi-API benchmark (Torchdata-Github) and a newly constructed benchmark (Torchdata-Manual) that involves more complex API interactions. Experimental results demonstrate that ExploraCoder significantly improves performance for models lacking prior API knowledge, achieving an absolute increase of 11.24% over naive RAG approaches and 14.07% over pretraining methods in pass@10. Moreover, we find that the integration of a self-debug mechanism in CoAE further boosts ExploraCoder's performance on more challenging tasks. Comprehensive ablation and case studies are conducted to provide further insights into the effectiveness of ExploraCoder.

## 2 RELATED WORK

**Code Generation with LLM.** Code generation, the process of automatically producing program code from user specifications, has seen remarkable advancements with the advent of LLMs (Guo et al., 2024; Qwen, 2024; OpenAI., 2024). By training on vast code corpora, these models have demonstrated a profound ability to generate contextually accurate and executable code snippets from high-level descriptions.

Recent research has increasingly focused on enhancing LLM performance in handling complex coding tasks. A line of work attempts to adopt the chain-of-thought (Wei et al., 2022) approach from the NLP community, where LLMs are prompted to generate intermediate reasoning steps before arriving at the final answer. However, Self-Planning(Jiang et al., 2024) argues that the original idea of CoT does not reduce the difficulty of coding problems, and advocates for more effective planning mechanisms tailored specifically for code generation. Another line of work utilizes execution feedback to improve generated code(Madaan et al., 2023). For instance, Self-Repair(Olausson et al., 2024) proposes a three-step framework that involves generating initial code, reflecting on error messages, and subsequently repairing the code based on those reflections.

**Library-Oriented Code Generation.** Real-world programming problems often involve the use of external libraries, posing a challenge for LLM when tasked to invoke APIs that were not seen in the training data. An intuitive solution to this issue is to pretrain or continue pretrain on the new API data(Zan et al., 2022). However, the training process is often computationally expensive and time consuming, making this approach difficult to deploy in practice.

An alternative approach leverages RAG techniques. Most previous studies adopt a naive RAG framework, where a retriever directly retrieves relevant APIs and provides them as context for final solution generation. However, Zan et al. (2024) observed that such a framework struggles with complex problems requiring multiple API invocations. Some recent work puts efforts into improving existing RAG frameworks. For examples, Ma et al. (2024b) proposed a decomposed retrieval method, incorporating an inter-task LLM reranker to suggest k APIs for code generation. However, their approach primarily focuses on API retrieval, and its performance in generating multiple API invocations remains limited. Li et al. (2024) fine-tuned a task decomposer using proprietary enterprise data and integrated RAG with CoT techniques by putting programming subtasks in context as step-by-step instructions for the generator. However, their evaluation relies solely on lexical and human scoring metrics, leaving the execution performance of their method unexplored.

**Unseen Library Benchmarks.** Constructing unseen library benchmarks[2] is particularly challenging due to limited resources and the difficulty of verifying whether the knowledge was included in model training. Previous work has generally involved manually building small-scale benchmarks for mono-library invocation tasks. For example, Zan et al. (2023) lexically transformed API names in public library programming problems to build MonkeyEval and BeatNumEval. But the API functionalities are already exposed to LLMs. Zan also developed TorchdataEval with 50 simple API invocation problems. However, these benchmarks only address basic programming problems mostly involving 1-2 API invocations. Ma et al. (2024b) proposed a more advanced Torchdata benchmark, featuring 50 programming tasks adapted from GitHub client code, each involving 3–8 API invocations. But the benchmark remain relatively simple, as many tasks focus on basic file reading operations with 3–4 API calls, whereas real-world software development often involves far more complex API interactions(Kula et al., 2018; Bauer et al., 2012). This gap highlights the need for a more comprehensive and complex unseen library benchmark.

## 3 EXPLORACODER FRAMEWORK

### 3.1 TASK DEFINITION

This work targets to addressing the task of library-oriented code generation (Zan et al., 2022), where the goal is to utilize library APIs to solve a programming problem. Formally, given a problem $\psi$ that specifies the user requirement and a set of API documents $\mathcal{A}$, a model $\theta$ generates solutions $p \sim \mathcal{P}_\theta(.|\psi, A)$.

---

[2]Also referred to as private library benchmark in some work

Most code libraries provide essential information such as API signatures, descriptions, library overviews, and example usage code. In this paper, we assume the accessibility of this information from the provided documents. As shown in Figure 1, ExploraCoder will automatically identify relevant subset of APIs $\hat{\mathcal{A}}$ and accumulate useful invocation experience $\hat{\mathcal{E}}$, which are then used as augmenting signals to enhance the code generation process:

$$p := ExploraCoder(\psi, \hat{\mathcal{A}}, \hat{\mathcal{E}}) \tag{1}$$

## 3.2 PLANNING FOR API INVOCATION

Real-world programming problems often involve multiple composite operations (Yu et al., 2024), necessitating a plan for **where** and **how** APIs can contribute to problem-solving. Specifically, we need to outline several simple API invocation subtasks, upon which ExploraCoder will sequentially explore the correct API calls. Ideally, we aim to set the planning granularity to simple subtasks where each requires only 1–2 API invocations to be completed. Excessive granularity can make subtasks overly challenging, while incorrect segmentation—such as planning subtasks that cannot be addressed by any of the library APIs—can lead the model to generate numerous hallucinations along the line (Liu et al., 2024a; Tian et al., 2024).

However, the granularity and functional boundaries of library APIs are domain-specific, often falling out of distribution (OOD) of LLMs when the library is absent from the model's training data. This misalignment poses a challenge in coordinating task planning with the typical usage patterns of these APIs.

To address this, we leverage the in-context learning capabilities of LLMs (Bareiß et al., 2022; An et al., 2023) by providing a condensed library overview and a small number of planner examples. This enables the LLMs to learn high-level usage patterns of the library. In this work, we prompt GPT-3.5-turbo-0125 to automatically summarize a piece of text $s$ from the library overview and extract few-shot planners $\mathcal{D} = \{\langle \psi_j, \{t_u\}_{u=1}^{w_j}\rangle\}_{j=1}^{n_\mathcal{D}}$ from the provided code examples, where $\psi_j$ is the requirement description of the $j$-th code example, and $t_u$ is the explanation of $u$-th API invocation. Note that we do not leak any detailed API usage or benchmark-related knowledge to models (detailed in Appendix A.6). Now, we can plan $n$ API invocation subtasks for a given problem $\psi$:

$$\{t_i\}_{i=1}^{n} \sim \mathcal{P}_\theta(.|\psi, \mathcal{D}, s) \tag{2}$$

## 3.3 API RECOMMENDATION

The API recommendation module serves to recommend relevant API documents $\mathcal{A}_i = \{a^{(1)}, \ldots, a^{(k)}\}$ for each API invocation subtask $t_i$. The API documents are transformed into a tabular style and then serve as the candidates for retrieval, where each row consists of the API import path, signature, and description. We first use a dense retriever to retrieve a initial subset of APIs by computing the similarity between $t_i$ and each $a_j$. More discussion of this module is provided in Appendix A.4.

$$\mathcal{A}_i = \text{top-}k \{\text{sim}(a_j, t_i) \mid a_j \in \mathcal{A}\} \tag{3}$$

Then, we prompt LLM to re-rank and drop irrelevant ones from the APIs retrieval results for each subtask, providing the refined set $\{\tilde{\mathcal{A}}_i\}_{i=1}^{n}$ for Chain of API Exploration, where then the actually used APIs $\tilde{\mathcal{A}}_{\text{CoAE}}$ will be recorded. Meanwhile, we also conduct an inter-task reranking (Ma et al., 2024b) to recommend a subset $\tilde{\mathcal{A}}_G$ with the volume of $k_G$ APIs from a global perspective. In the final solution stage, we provide for the generator:

$$\hat{\mathcal{A}} = \tilde{\mathcal{A}}_{\text{CoAE}} \cup \tilde{\mathcal{A}}_G \tag{4}$$

## 3.4 CHAIN OF API EXPLORATION

After preparing $n$ API invocation subtasks $\{\langle \tilde{\mathcal{A}}_i, t_i \rangle\}_{i=1}^{n}$ through the previous two modules, the next step is to find the correct APIs and their appropriate usage for each subtask.

Previous work shows LLMs struggle to directly invoke multiple unseen APIs in a single run to solve a complex problem (Zan et al., 2024). The difficulty arises because LLMs tend to hallucinate the

usage of unfamiliar APIs, and errors can propagate through APIs interactions, further compounding the issue. In contrast, when lacking knowledge of relevant APIs, programmers could apply an exploratory programming style, experimenting with code in a sandbox environment to accumulate experience about correct API usage. Inspired by this behaviour, we designed a Chain of API Exploration to help LLMs autonomously explore API usage and sequentially solve the tasks related to the problems. We now formalize the main steps in CoAE.

**Experimental code generation.** We encourage the LLM to generate $m$ diversified experimental code snippets for subtask $t_i$:

$$\{p_{i,j}\}_{p_{j=1}}^m \sim \mathcal{P}_\theta(.|t_i, s, \hat{\mathcal{A}}_i, \mathcal{E}_{1:i-1}) \tag{5}$$

where $s$ is the high-level library information from Section 3.2, and $\mathcal{E}_{1:i-1}$ is the accumulated invocation experience from subtasks prior to $t_i$. We define API invocation experience as the combination of a code snippet and its observed execution output, which will be elaborated in the next step. Each experimental code will attempt to solve the subtask by making different API invocations, and print out important information such as API's returned object. Such feedback will be observed by LLM in the next step.

In our experiments, we use the example inputs provided by the programming problem as a hint, and prompt LLM to reuse or create its own test input for experiments. For a subtask that requires interactions with prior subtasks, the LLM adapts the code snippet from prior invocations to construct inputs for the currect API invocation.

**Code execution and observation.** A line of work execute candidate codes against prepared test cases to facilitate code ranking (Shi et al., 2022) or refinement (Zhang et al., 2023). However, quality testbeds are often inaccessible in production environments (Siddiq et al., 2023; Schäfer et al., 2023), particularly when developing new functionalities using newly introduced or unfamiliar libraries. Furthermore, creating test cases at runtime for the newly planned subtasks is even more challenging. Instead, we directly execute the experimental codes in a sandbox environment and capture its output. Specifically, given $t_i$ and $p_{i,j}$, the observation $o_{i,j}$ by the LLM consists of the codes' executability $\delta$, error message $\varepsilon$, and program output $\gamma$. We now can assemble $m$ candidate experience for $t_i$ as:

$$\mathcal{E}_i = \{\langle t_i, p_{i,j}, o_{i,j}\rangle\}_{j=1}^m \tag{6}$$

**Enhance experience exploitation by self-debugging.** In our preliminary experiments, we found the experimental codes often fail to execute due to simple mistakes (e.g., missing import statements). Additionally, some challenging tasks require complex interactions with APIs from prior subtasks, which LLMs struggle to solve without execution feedback. preventing the observation of valuable insights into API behavior. And these failures could propagate along the exploration chain.

To address this, we prompt the LLM to autonomously repair failed codes when all candidate codes for a given task fail to execute, thereby enriching its execution experience. We report the effectiveness of ExploraCoder, both with and without self-debugging capabilities in Section 5.

**Experience Selection Strategy.** We have obtained $m$ candidate exploration experience $\{\mathcal{E}_{i,j}\}_{j=1}^m$ on $t_i$. The goal in this step is to select the most valuable experience $\hat{\mathcal{E}}_i$ and prune the others for $t_i$. In this work, we adopt a simple selection strategy: (1) randomly select an experience candidate that has successfully executed; (2) if all candidates fail to execute, we randomly select a failed one. Then, the selected experience is placed into the exploration chain, which will be passed on to the next subtask and accumulates progressively. Ultimately, we obtain a experience trace of the following form to aid in solution generation:

$$\hat{\mathcal{E}} = \{\hat{\mathcal{E}}_i\}_{i=1}^n \tag{7}$$

## 4 BENCHMARK CONSTRUCTION

Unseen library benchmarks provide essential references for evaluating LLMs' performance in handling unseen APIs. Real-world programming often involves multiple unseen APIs. However, existing benchmarks typically provide problems with only 1-2 APIs. To address this limitation, we con-

Table 1: Statistical Summary of two Torchdata-based benchmarks. Num. APIs. and Avg. APIs. report the range and average number of distinct APIs involved in each sample. Num. Invoc. and Avg. Invoc. report the range and average number of API invocations in the samples' canonical solution. Volume of the doc pool refers to the number of API documents provided by the library, which also represents the size of the search space during API retrieval.

| Benchmarks | Num. samples | Num. APIs | Avg. APIs | Num. Invoc. | Avg. Invoc. | Volume of doc pool |
|---|---|---|---|---|---|---|
| Torchdata-Github | 50 | 3-8 | 4.26 | 3-8 | 4.64 | 228 |
| Torchdata-Manual | 50 | 8-14 | 9.94 | 8-21 | 12.00 | 228 |

structed two Torchdata-based[3] multi-API benchmarks: Torchdata-Github and Torchdata-Manual. Detailed statistics are presented in Table 1.

**Torchdata-Github.** We curated the 50 Torchdata problems provided by Ma et al. (2024b) and constructed the Torchdata-Github benchmark. These programming problems are adapted from client project of Torchdata on GitHub, each containing a coarse-grained user requirement that entails 3-8 API invocations. Specifically, we add example inputs for each problem to facilitate CoAE. And we manualy supplemented external resources needed to run test cases in some problems[4]

**Torchdata-Manual.** To evaluate the models' ability to address more complex API invocations, we developed a new benchmark called Torchdata-Manual, comprising 50 manually crafted programming problems. Each problem involves 8-14 distinct Torchdata APIs and more clearly stated problem descriptions. To ensure the diversity of the programming tasks, we randomly sampled numerous API combinations from the Torchdata documentation pool and selected plausible combinations to formulate the problem. Two programmers with more than five years of Python coding experience are invited to review the benchmark. More detailed construction methodology is provided in the Appendix A.5.

## 5 EXPERIMENTS

### 5.1 EXPERIMENTAL SETUPS

**Benchmarks and base language models.** We evaluate ExploraCoder's performance on multi-API invocation problems using the Torchdata-Github and Torchdata-Manual benchmarks. Based on the the publicly available information on models' training data cutoff date, we conduct our main experiments under two base models settings: (1) *API-untrained model*: where the API knowledge is unseen by model during training phase. We choose GPT-3.5-turbo-0125 and GPT-4-0613[5] as representatives. (2) *API-pretrained model*: where the API knowledge is pretrained in model. We represent it by a newer version of GPT-4-1106-preview and two SOTA opensource code LLM: CodeQwen-1.5 and DeepseekCoder-6.7b[6]. Due to the token budgets and inference speed, we primarily experiment ExploraCoder with GPT-3.5-turbo-0125, while reporting GPT-4-0613 results where necessary to further support our conclusions.

**Evaluation metrics.** We adopt **Pass@k** as our primary evaluation metrics in accordance with previous work (Zan et al., 2023). For each problem, we randomly sample $n \geq k$ code solutions from the model to execute against test cases. And pass@k is calculated as the percentage of problems solved using k candidates per problem.

---

[3]Torchdata is released in May 2022. The LLMs used in this study (GPT-3.5-turbo and GPT-4-0613) were pretrained on GitHub corpora before Torchdata's release, meaning the APIs were unseen by them during training. Moreover, since its release, Torchdata has accumulated open-source client code, allowing more recently trained LLMs to acquire knowledge of Torchdata. The choice of Torchdata library provides an opportunity to compare our training-free framework with models containing pretrained knowledge.

[4]Some external resources, such as local files to be loaded in problems, are not provided by Ma et al. (2024b).

[5]See gpt-3.5-turbo and gpt-4 on `https://platform.openai.com/docs/models/overview`

[6]We use the instruct-tuned version of code LLM. See `https://qwenlm.github.io/blog/codeqwen1.5/` and `https://deepseekcoder.github.io/`

Table 2: Performance comparison on Torchdata-Github. We compare with SOTA approach in later section.

| API Knowledge | Method | $k = 1$ | | $k = 5$ | | $k = 10$ | | $k = 20$ | |
|---|---|---|---|---|---|---|---|---|---|
| | | Pass | Success | Pass | Success | Pass | Success | Pass | Success |
| Pretrained in models | DeepSeekCoder-6.7B | 5.24% | 6.86% | 14.43% | 19.28% | 18.64% | 27.38% | 21.80% | 37.23% |
| | CodeQwen1.5-7B | 3.24% | 6.10% | 11.60% | 19.94% | 16.57% | 28.56% | 19.90% | 37.42% |
| | GPT-4-1106-preview | 7.43% | 11.52% | 16.19% | 28.88% | 21.34% | 38.74% | 25.81% | 45.71% |
| Untrained in models | GPT-3.5-turbo-0125 | 1.70% | 2.09% | 5.54% | 6.95% | 7.28% | 9.64% | 8.00% | 11.90% |
| | + naive RAG | 6.00% | 10.57% | 10.55% | 24.00% | 14.67% | 32.50% | 20.83% | 40.81% |
| | **+ ExploraCoder** | **10.19%** | **19.50%** | **18.64%** | **39.39%** | **21.67%** | **48.56%** | **25.62%** | **57.30%** |
| | GPT-4-0613 | 3.50% | 5.43% | 8.86% | 16.35% | 11.45% | 23.79% | 13.80% | 31.52% |
| | + naive RAG | 10.09% | **29.64%** | 20.11% | 39.04% | 24.07% | 45.16% | 27.81% | 49.33% |
| | **+ ExploraCoder** | **15.43%** | 23.10% | **21.53%** | **45.62%** | **28.11%** | **55.25%** | **30.00%** | **61.87%** |

Table 3: Performance comparison on Torchdata-Manual. We compare with SOTA approach in later section

| API Knowledge | Method | $k = 1$ | | $k = 5$ | | $k = 10$ | | $k = 20$ | |
|---|---|---|---|---|---|---|---|---|---|
| | | Pass | Success | Pass | Success | Pass | Success | Pass | Success |
| Pretrained in models | DeepSeekCoder-6.7B | 0% | 0.48% | 0% | 1.57% | 0% | 1.95% | 0% | 2.00% |
| | CodeQwen1.5-7B | 0% | 0.39% | 0% | 1.43% | 0% | 2.86% | 0% | 5.71% |
| | GPT-4-1106-preview | 0.19% | 1.33% | 0.95% | 6.15% | 1.90% | 11.41% | 3.80% | 21.05% |
| | + naive RAG | 1.43% | 3.90% | 5.87% | 15.32% | 9.51% | 23.03% | 13.90% | 27.90% |
| | + ExploraCoder | 9.91% | 29.86% | 23.66% | 51.76% | 29.30% | 58.19% | 33.71% | 63.10% |
| Untrained in models | GPT-3.5-turbo-0125 | 0% | 0% | 0% | 0% | 0% | 0% | 0% | 0% |
| | + naive RAG | 0.10% | 0.48% | 0.48% | 2.20% | 0.95% | 3.90% | 1.90% | 5.90% |
| | **+ ExploraCoder** | **4.48%** | **9.71%** | **8.71%** | **18.60%** | **11.61%** | **22.45%** | **13.79%** | **25.80%** |
| | GPT-4-0613 | 0% | 0% | 0% | 0% | 0% | 0% | 0% | 0% |
| | + naive RAG | 0% | 0.57% | 0% | 2.59% | 0% | 4.61% | 0% | 7.71% |
| | **+ ExploraCoder** | **10.85%** | **21.24%** | **19.59%** | **35.73%** | **23.27%** | **40.87%** | **27.71%** | **45.70%** |

In our preliminary study with Torchdata-Manual, we find that some baselines cannot pass any programming problem at all. To better observe their nuance performance differences, we additionally report **Success@k** (Chen et al., 2024) in experiments. Success rate relaxes the evaluation criteria by measuring whether the generated code can be executed successfully without runtime errors within limited timeout constraints.

**Implementation details.** We implement ExplorCoder by setting $k_{\mathcal{D}} = 5$ for task planning module. For API recommendation, we set $k = 20$ as initial retrieval volume, and we set $k_G = 15$ on Torchdata-Github following Ma et al. (2024b) and $k_G = 20$ on Torchdata-Manual. For CoAE, we set $m = 5$. To generate diverse candidates, we set the $temperature = 0.8$ and $top\_p = 0.95$ for our CoAE and for final solution generation across all baselines. More detailed experimental settings are left in Appendix A.6

## 5.2 Generating multiple API invocations with LLMs

Firstly, we analyze the effectiveness of different knowledge injection approaches in library-oriented code generation tasks in Table 2 and 3.

**Invoking APIs using API-untrained and API-pretrained models.** By analyzing the direct generation performance of the five base models, we observe that API-pretrained models consistently outperform API-untrained models. This highlights the importance of prior API knowledge in library-oriented code generation. And the lower performance across all models on the Torchdata-Manual (Table 3) further underscores the challenge posed by more complex API invocations, making it a more effective benchmark for our task. [7]

---

[7]Notably, API-untrained models manage to solve a small number of problems in Torchdata-Github even without explicit API information in the context. We find in Appendix A.7 that these models circumvent the lack of API knowledge by invoking built-in APIs that provide similar functionalities and meet the task requirements.

Table 4: Performance comparison between ExploraCoder and ExploraCoder* on Torchdata-Manual.

| Benchmark | Model | Method | k = 1 | | k = 5 | | k = 10 | | k = 20 | |
|---|---|---|---|---|---|---|---|---|---|---|
| | | | Pass | Success | Pass | Success | Pass | Success | Pass | Success |
| Torchdata-Github | GPT-3.5-turbo | ExploraCoder | 10.19% | 19.50% | 18.64% | 39.39% | 21.67% | 48.56% | 25.62% | 57.30% |
| | | **ExploraCoder*** | **19.24%** | **38.66%** | **25,41%** | **54.93%** | **27.64%** | **59.56%** | **31.62%** | **63.71%** |
| | GPT-4-0613 | ExploraCoder | 15.43% | 23.10% | 21.53% | 45.62% | **28.11%** | 55.25% | 30.00% | 61.87% |
| | | **ExploraCoder*** | **19.52%** | **40.19%** | **25.50%** | **58.78%** | 27.66% | **68.05%** | **31.18%** | **70.37%** |
| Torchdata-Manual | GPT-3.5-turbo | ExploraCoder | 4.48% | 9.71% | 8.71% | 18.60% | 11.61% | 22.45% | 13.79% | 25.80% |
| | | **ExploraCoder*** | **8.76%** | **15.48%** | **14.04%** | **25.31%** | **16.75%** | **29.91%** | **19.81%** | **33.81%** |
| | GPT-4-0613 | ExploraCoder | 10.85% | 21.24% | 19.59% | 35.73% | 23.27% | 40.87% | 27.71% | 45.70% |
| | | **ExploraCoder*** | **15.24%** | **30.00%** | **29.77%** | **54.58%** | **36.22%** | **62.71%** | **41.70%** | **69.52%** |

Through a naive RAG framework (Zhou et al., 2023), the performance of API-untrained models has been effectively improved, bridging the gap caused by the lack of API knowledge compared to API-pretrained models. Specifically, GPT-3.5-turbo-0125 + naive RAG achieves 7.36% absolute increase in pass@20 compare to its base model across two datasets. We make a fairer comparison of RAG methods and pretraining methods by looking into two GPT-4 models. We discussed the fairness in Appendix A.2. GPT-4-0613 + naive RAG outperforms GPT-4-1106-preview by an average of 2.69% absolute increase in pass@k on Torchdata-Github, but underperforms on the more challenging Torchdata-Manual benchmark.

**ExploraCoder vs naive RAG on API-untrained models.** From Table 2 and 3, we can observe that ExploraCoder brings substantial improvements over naive RAG for both two API-untrained models (GPT-3.5-turbo-0125 and GPT-4-0613), with an average absolute gains in pass@20 of 3.5% on Torchdata-Github and 19.8% on Torchdata-Manual. These improvements could be attributed to ExploraCoder's potential in addressing two limitations of the naive RAG framework when handling complex API invocation subtasks:

(1) *Retrieval for complex requirement*: In the naive RAG approach, the retriever's ability to recall relevant APIs for comprehensive requirements becomes a bottleneck (Please refer to Appendix A.4). ExploraCoder addresses this by using a divide-and-conquer strategy, retrieving APIs for each individual invocation subtask. Additionally, ExploraCoder alleviates dependency on the retrieval number hyperparameter by setting a fixed API number for simple subtasks and dynamically adjusting the subtask number.

(2) *Generating code with multiple unseen APIs*: Current models struggle to generate code that invokes multiple unseen APIs due to the cognitive complexity involved in understanding new APIs and managing their interactions (Please refer to Appendix A.7 for case study). ExploraCoder mitigates this challenge by adopting a human-like exploratory programming paradigm, where it incrementally generates simple, reusable code snippets that are highly relevant to problem solving during CoAE, and learns the API usage knowledge by observing the API interactions behavior.

**ExploraCoder on API-pretrained model.** We observe that the API-pretrained GPT-4-1106-preview underperforms on Torchdata-Manual, achieving a pass@1 of only 0.19%. Therefore, we use GPT-4-1106-preview on Torchdata-Manual benchmark as a proxy to further examine the effectiveness of ExploraCoder on API-pretrained models. As shown in Table 3, ExploraCoder achieves a substantial improvement over GPT-4-1106-preview, with an absolute pass@1 increase of 9.72%, and it also outperforms GPT-4-1106-preview + naive RAG by 8.48%. These results indicate that ExploraCoder is universally effective, improving models with varying levels of pretraining on relevant API knowledge.

## 5.3 Boosting ExploraCoder with experience exploitation

Our quantitative analysis (Please refer to Appendix A.3) shows a positive correlation between CoAE success rate and the final solution success rate. In cases that have more successful executed subtasks, particularly those with a CoAE success rate of 1, are more likely to generate successful or passing final solutions. Intuitively, the failures in the exploration chain, such as wrong API usage could propagate through API interactions and affect the generation of final solutions. Therefore, we wonder if we can enhance ExploraCoder by improving the success rate in each API invocation subtask.

Table 5: Comparing ExploraCoder with baseline approaches using GPT3.5 on Torchdata-Github.

| Method | $k = 1$ | | $k = 5$ | | $k = 10$ | | $k = 20$ | |
|---|---|---|---|---|---|---|---|---|
| | Pass | Success | Pass | Success | Pass | Success | Pass | Success |
| Direct | 1.70% | 2.09% | 5.54% | 6.95% | 7.28% | 9.64% | 8.00% | 11.90% |
| DocPrompting (Zhou et al., 2023) | 6.00% | 10.57% | 10.55% | 24.00% | 14.67% | 32.50% | 20.83% | 40.81% |
| CAPIR (Ma et al., 2024b) | 5.90% | 10.47% | 14.59% | 27.08% | 18.60% | 37.19% | 23.52% | 47.43% |
| EpiGen (Li et al., 2024) | 8.57% | 18.95% | 14.63% | 35.61% | 17.24% | 41.67% | 19.61% | 47.62% |
| **ExploraCoder (Ours)** | **10.19%** | **19.50%** | **18.64%** | **39.39%** | **21.67%** | **48.56%** | **25.62%** | **57.30%** |
| Self-Repair(Olausson et al., 2024) | 16.47% | 22.10% | 21.04% | 29.70% | 21.75% | 32.20% | 22.00% | 33.90% |
| **ExploraCoder* (Ours)** | **19.24%** | **38.66%** | **25,41%** | **54.93%** | **27.64%** | **59.56%** | **31.62%** | **63.71%** |

Table 6: Comparing ExploraCoder with baseline approaches using GPT3.5 on Torchdata-Manual.

| Method | $k = 1$ | | $k = 5$ | | $k = 10$ | | $k = 20$ | |
|---|---|---|---|---|---|---|---|---|
| | Pass | Success | Pass | Success | Pass | Success | Pass | Success |
| Direct | 0% | 0% | 0% | 0% | 0% | 0% | 0% | 0% |
| DocPrompting (Zhou et al., 2023) | 0.10% | 0.48% | 0.48% | 2.20% | 0.95% | 3.90% | 1.90% | 5.90% |
| CAPIR (Ma et al., 2024b) | 2.76% | 3.81% | 5.17% | 9.51% | 5.79% | 12.77% | 6.00% | 15.81% |
| EpiGen (Li et al., 2024) | 1.62% | 5.73% | 3.16% | 13.44% | 3.75% | 17.00% | 3.97% | 21.43% |
| **ExploraCoder (Ours)** | **4.48%** | **9.71%** | **8.71%** | **18.60%** | **11.61%** | **22.45%** | **13.79%** | **25.80%** |
| Self-Repair(Olausson et al., 2024) | 5.33% | 14.19% | 6.84% | 18.30% | 7.48% | 19.66% | 7.99% | 20.00% |
| **ExploraCoder* (Ours)** | **8.76%** | **15.48%** | **14.04%** | **25.31%** | **16.75%** | **29.91%** | **19.81%** | **33.81%** |

To this end, we designed an enhanced ExploraCoder* by introducing an extra self-debug step into the CoAE. When the exploration tries reaches a threshold number and no candidate code is executable, instead of keep exploring more candidates codes, we enhance the exploitation of the existing candidates by prompting LLMs to debug on the failed ones. Table 4 shows that ExploraCoder* significantly boosts the final solution's quality, achieving an average increase of 50.33% in success@1 and 68% in pass@1.

## 5.4 COMPARING WITH RELATED APPROACHES

In this section, we further compare ExploraCoder with other advanced RAG-based approaches. We also include Docprompting, the previously reported naive RAG framework, along with direct generation method as baselines.

For CAPIR and EpiGen, we set a fixed number API recommendation in accordance with our $A_G$, and we directly use the subtasks generated by ExploraCoder's planning module as the CoT instruction for EpiGen. Tables 5 and 6 show that both CAPIR and EpiGen improve upon the naive RAG baseline, while ExploraCoder further surpasses these two approaches, with an relative increase of 82.6% on pass@1 across two benchmarks.

To enable a fair comparison with ExploraCoder*, we adapted a SOTA debugging framework, Self-Repair (Olausson et al., 2024), with retrieval augmentation. We incorporate CAPIR's API retrieval results into the LLM's context throughout Self-Repair's 3-stage generation process. We compare ExploraCoder and Self-Repair under the same computational budget. Specifically, for each problem, ExploraCoder generates $n$ plans, enabling up to $n$ debug operations in CoAE, we set the iteration budget for Self-repair to $n$ accordingly.

Table 5 and 6 shows that ExploraCoder outperforms self-repair by a significant margin of 40.6% on pass@1. While Self-Repair can effectively boost the success rate for CAPIR, the overall improvement in the pass rate remains limited. This limitation stems from the nature of next-token prediction models, where API usage hallucinations tend to accumulate throughout the API invocation sequence, particularly in subtasks involving multiple APIs. In many cases, we observe that Self-Repair repeatedly attempts to fix a buggy codes that deviates substantially from the correct solution, continuing until it exhausts its repair budget (Please refer to Appendix A.7 for case study). In contrast, ExploraCoder mitigates this issue by leveraging timely execution feedback during simpler early-stage tasks, using this feedback to inform and generate accurate API invocations. Additionally, Self-Repair demonstrates significant performance gains when considering a small number of

Table 7: Ablation study for GPT3.5 + ExploraCoder* on Torchdata-Manual.

| Method | $k = 1$ | | $k = 5$ | | $k = 10$ | | $k = 20$ | |
|---|---|---|---|---|---|---|---|---|
| | Pass | Success | Pass | Success | Pass | Success | Pass | Success |
| ExploraCoder* | 8.76% | 15.48% | 14.04% | 25.31% | 16.75% | 29.91% | 19.81% | 33.81% |
| w/o self-debug | 4.48% | 9.71% | 8.71% | 18.60% | 11.61% | 22.45% | 13.79% | 25.80% |
| w/o domain knowledge | 3.90% | 15.22% | 6.86% | 26.83% | 7.90% | 30.79% | 9.82% | 35.62% |
| w/o CoAE | 0.85% | 1.81% | 3.38% | 7.19% | 5.33% | 11.59% | 7.30% | 17.52% |
| w/o experience selection | 5.81% | 14.33% | 9.53% | 25.11% | 9.98% | 26.87% | 10.00% | 27.90% |

output candidates. However, its relative improvement decreases as k increases. This is because the debugging process in Self-Repair relies on explicit and consistent feedback, leading to highly similar solutions. ExploraCoder can better harness the potential of output candidates by utilizing a standalone solution generator.

## 5.5 ABLATION STUDY

We further conducted an ablation study on our best-performing framework, ExploraCoder*, in Table 7. We experiment on the Torchdata-Manual benchmark using GPT-3.5-turbo-0125. As discussed earlier, self-debug can effectively improve the performance of ExploraCoder. In practical deployment, users may consider incorporating more debugging iteration in CoAE to further boost the performance of ExploraCoder.

We then ablate the library-level domain knowledge (few-shot planner $\mathcal{D}$ and library introduction $s$) provided to ExploraCoder, and let the model plan API invocation subtasks based on its commonsense knowledge. We found that its execution success rate is comparable to that of ExploraCoder*, but the pass rate decreases. The model often breaks down problem into overly coarse-grained tasks that are difficult to solve, leading to buggy codes early in the chain. During the process of debugging and exploring along the chain, the LLM gradually drifts away from the original problem. Although it eventually produces executable code based on biased experience, the resulting program fails to fully meet the requirements.

We also ablate the CoAE by providing all the retrieved API information throughout ExploraCoder's process to the generator, and prompt it to generate solution in a single run. We find that the performance significantly drop to 0.85 in pass@1. This suggests that (1) the current SOTA models still lack adequate in-context reasoning ability to handle complex tasks involving multiple API invocations, and (2) the information provided in API documentation is often insufficient, leading to hallucinations in API usage generation. This highlights the need for execution feedback to clarify ambiguities in API usage.

We further ablate a critical step within CoAE by removing the experience selection process. Instead of using the current selection strategy, we random select the candidates, ignoring their execution feedback. We find that the model still perform reasonably well on pass@1. However, as $k$ increases, its performance quickly converge to its limitation. We suggest this is because the model did not effectively leverage candidate experience information. An unexecutable API invocation experience provides limited insights to the API usage, and may even introduce some harmful effects to the final solution generation.

## 6 CONCLUSION

We present ExploraCoder, a novel code generation framework for LLMs to generate codes with multiple unseen API invocations through planning API invocation tasks and experiments with each task in a chain of API Exploration. We construct two challenging benchmarks Torchdata-Github and Torchdata-Manual to evaluate LLM's performance when dealing with complex requirements that involve multiple unseen API invocation. Our experiments present the significant performance improvement of ExploraCoder when comparing with both related RAG frameworks and pretraining appraoches. We find that adding a self-debug component in the API exploration chain can fur-

ther boost the performance of ExploraCoder. We provide comprehensive ablation studies to clearly present the benefits obtained by each of the components.

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

## A  APPENDIX

### A.1  EVALUATING API KNOWLEDGE RETENTION IN API-PRETRAINED AND API-UNTRAINED BASE MODELS

We methodically differentiate between API-pretrained and API-untrained models based on publicly available information regarding their training data. However, to intuitively investigate whether the model retains substantial knowledge of Torchdata APIs, we directly ask the model to provide specific API details, as shown in Listing 1. We set $temperature = 0$ and $top\_p = 1$ across the models. Our findings reveal that API-untrained models (GPT-3.5-turbo-0125 and GPT-4-0613) hallucinate API information, generating API names that do not exist in the Torchdata library. In contrast, the API-pretrained models (GPT-4-1106-preview, CodeQwen1.5-7B-Chat, deepseek-coder-6.7b-instruct) accurately generates correct API names, import paths, and even provides precise usage descriptions. These results provide evidence supporting the validity of our experimental settings.

Listing 1: Demonstration of API knowledge in LLMs.

```
what are some APIs in torchdata? Give me 5 specific API names(including full import path)
--------- output by GPT-3.5-turbo-0125 ---------
1. torchdata.datasets.ImageFolderDataset
   - Full import path: torchdata.datasets.ImageFolderDataset
2. torchdata.datasets.CIFAR10Dataset
   - Full import path: torchdata.datasets.CIFAR10Dataset
3. torchdata.datasets.MNISTDataset
   - Full import path: torchdata.datasets.MNISTDataset
4. torchdata.transforms.Compose
   - Full import path: torchdata.transforms.Compose
5. torchdata.transforms.RandomHorizontalFlip
   - Full import path: torchdata.transforms.RandomHorizontalFlip
--------- output by GPT-4-0613 ---------
1. torchdata.datasets.Files: This API is used to create a dataset from a list of files. It can be used to
   load and preprocess data from different file formats.
2. torchdata.datasets.Dataset: This is the base class for all datasets in torchdata. It provides methods for
   loading and iterating over data.
3. torchdata.datasets.Transform: This API is used to apply transformations to the data in a dataset. It can
   be used to preprocess data, such as normalizing or augmenting it.
4. torchdata.datasets.Folder: This API is used to create a dataset from a directory of files. It can be used
   to load and preprocess data from different file formats.
5. torchdata.datasets.ImageFolder: This API is used to create a dataset from a directory of images. It can be
   used to load and preprocess image data.
--------- output by GPT-4-1106-preview ---------
1. `torchdata.datapipes.iter.FileOpener`:
   ```python
   from torchdata.datapipes.iter import FileOpener
   ```
   This API is used to open files from a given source (like a list of file paths) and yield a tuple of the
   file name and a file object.
2. `torchdata.datapipes.iter.HttpReader`:
   ```python
   from torchdata.datapipes.iter import HttpReader
   ```
   This API allows you to read data from HTTP URLs. It takes an iterable of URLs and yields the contents of
   the URLs.
3. `torchdata.datapipes.iter.CSVParser`:
   ```python
   from torchdata.datapipes.iter import CSVParser
   ```
   This API is used to parse CSV files. It takes an iterable of tuples (usually coming from `FileOpener`) and
   yields the parsed CSV content.
4. `torchdata.datapipes.iter.RoutedDecoder`:
   ```python
   from torchdata.datapipes.iter import RoutedDecoder
   ```
   This API is a decoder datapipe that applies different decoders based on the extension of the file path or
   a custom function provided by the user.
5. `torchdata.datapipes.iter.Batcher`:
```

Table 8: We compare the success rate of ExploraCoder on two datasets.

| Base model | Dataset | length of CoAE | CoAE Success | Solution Success |
|---|---|---|---|---|
| GPT-3.5-turbo | Torchdata-Github | 4.06 | 63.00% | 57.30% |
| | Torchdata-Manual | 8.22 | 42.19% | 33.81% |
| GPT-4-0613 | Torchdata-Github | 5.28 | 71.74% | 61.87% |
| | Torchdata-Manual | 8.80 | 57.60% | 45.70% |

```python
from torchdata.datapipes.iter import Batcher
```
This API batches data points into a specified batch size. It's useful for creating mini-batches of data
    for training machine learning models.
--------- Output by CodeQwen1.5-7B-Chat ---------
Here are five specific API names in torchdata, along with their full import paths:
1. `torchdata.datapipes.iter.IterDataPipe`: This is the base class for all iterable data pipes in torchdata.
2. `torchdata.datapipes.iter.Mapper`: This data pipe applies a function to each element in the input data
    pipe.
3. `torchdata.datapipes.iter.Filter`: This data pipe filters out elements from the input data pipe based on a
    condition.
4. `torchdata.datapipes.iter.Shuffler`: This data pipe shuffles the elements in the input data pipe.
5. `torchdata.datapipes.iter.Batcher`: This data pipe batches elements from the input data pipe into
    fixed-size groups.
--------- Output by deepseek-coder-6.7b-instruct ---------
TorchData is a package in PyTorch that provides data utilities and primitives to create complex data
    pipelines. Here are five specific APIs in torchdata:
1. `torchdata.datapipes.iter.FileLister`: This API is used to list files in a directory.
2. `torchdata.datapipes.iter.FileOpener`: This API is used to open files.
3. `torchdata.datapipes.iter.LineReader`: This API is used to read lines from a file.
4. `torchdata.datapipes.iter.CSVParser`: This API is used to parse CSV files.
5. `torchdata.datapipes.iter.MapDataPipe`: This API is used to apply a function to each element in the data
    pipe.
Please note that these are just a few examples. The full list of APIs in torchdata can be found in the
    [official documentation](https://pytorch.org/data/).
```

## A.2 DISCUSSION OF FAIRNESS COMPARISON BETWEEN GPT-4-0613 AND GPT-4-1106-PREVIEW.

GPT-4-0613 and GPT-4-1106-preview are two closely released version of GPT-4. According to publicly available information, the former is trained on data available up until September 2021, while the latter is a more recent version trained on data up until April 2023. In our experiment, we assume that both models share a similar architecture , and that the performance gap of direct generation between the two is primarily due to the absence of API knowledge in training corpura, i.e. the performance gap between API-pretrianed and API-untrained models. Appendix A.1 has shown that while GPT-4-0613 is unaware of the Torchdata APIs, GPT-4-1106 can effectively recite the API details. In this context, we demonstrate in Section 5.2 that integrating our ExploraCoder framework allows API-untrained models to surpass their API-pretrained counterparts, whereas integrating naive RAG does not, proving the effectiveness of ExploraCoder.

## A.3 QUANTITATIVE ANALYSIS ANALYSIS FOR COAE

ExploraCoder leverages API invocation experience from CoAE to enhance the quality of final solution generation. Intuitively, the quality of exploration subtasks within CoAE is closely related to the quality of the final solutions. We first examine the overall success rate of CoAE subtasks and final solutions in ExploraCoder in Table 8. We observe that ExploraCoder's performance declines on Torchdata-Manual that involves longer exploration chains, with a 17.48% lower CoAE subtask success rate and a 19.83% lower final solution success rate comparing to Torchdata-Github. This could be attributed to failures in the exploration chain propagating to the generation of final solutions.

To further explore this relationship, we conducted a quantitative analysis, examining how the number of CoAE subtasks and their success rates affect the pass rate and overall success rate of the final solutions. We illustrate the correlation in a scatter plot (Figure 2), using results from the best-performing base model, GPT-4-0613, on our Torchdata-Manual benchmark. This dataset was chosen due to its diverse range of API invocation complexities, which result in varied numbers of decomposed subtasks, providing better visualization of the relationship. From Figure 2a to 2d, we observe that both the pass rate and success rate of the final solutions positively correlate with the

Table 9: Choices of hyperparameters for ExploraCoder's Retrieval module: we tested retrieval modules effectiveness by 3 aspects: choice of embedding model, retrieval index(Desc: API description, Path: API import path, Desc*: truncated first sentense of API description), retrieval method(ST: single-task, MT: multi-task).

| Retreival Model | Retrieval Index | Retrieval Method | Racall@3 | Recall@5 | Recall@10 | Revall@15 |
|---|---|---|---|---|---|---|
| BM25 | Desc | ST | 10.23 | 13.02 | 18.07 | 23.12 |
| BGE | Desc | ST | 12.57 | 16.50 | 25.49 | 32.81 |
| ADA | Desc | ST | 13.24 | 17.88 | 28.83 | 36.18 |
| ADA | Path | ST | 10.91 | 13.46 | 16.48 | 19.75 |
| ADA | Desc* | ST | 16.00 | 22.40 | 32.66 | 40.21 |
| ADA | Path+Desc* | ST | 19.64 | 24.66 | 34.28 | 40.91 |
| ADA | Path+Desc* | MT | 24.89 | 31.25 | 43.63 | 52.07 |

**CoAE subtask success rate.** Subtasks with higher success rates, particularly those with a success rate of 1, are more likely to generate successful or passing final solutions. Interestingly, the number of subtasks doesn't appear to have a significant direct impact. However, as shown in Figures 2a and 2b, without self-debugging, problems with a higher subtask number (ranging from 10 to 13) tend to have lower subtask success rate as the subtask number increase. This may be due to the increased complexity of inter-task API interactions, which can overwhelm LLMs. When the self-debug mechanism is introduced in ExploraCoder*, we observe in Figures 2c and 2d a notable improvement in the overall subtask success rate, even for cases with higher subtask numbers. This leads to more successful and passing final solutions. The improvement can be attributed to ExploraCoder's ability to correct typos and simple API interaction errors in each subtask, thereby gaining richer API usage experience and exploiting it to the final solution generation.

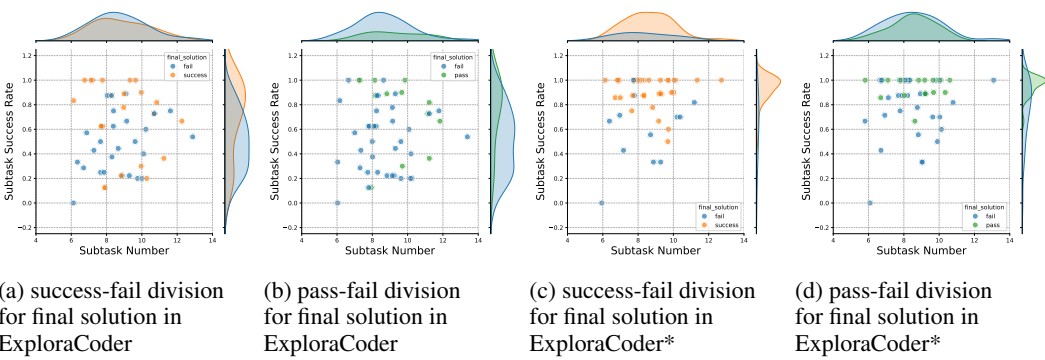

(a) success-fail division for final solution in ExploraCoder

(b) pass-fail division for final solution in ExploraCoder

(c) success-fail division for final solution in ExploraCoder*

(d) pass-fail division for final solution in ExploraCoder*

Figure 2: correlation between the quality of CoAE subtasks and final solutions

### A.4 MORE EXPERIMENTS ON API RECOMMENDATION MODULE IN EXPLORACODER

We evaluate the effectiveness of our API recommendation module using the Torchdata-AR benchmark (Ma et al., 2024b). Our experiments explored the performance difference in variant hyperparameters of the retrieval model, retrieval index, and retrieval methodology. Specifically, we assess the performance of a lexical method BM25 and two SOTA dense retrieval models, bge-large-en-v1.5, and text-embedding-ada-002. Our results show that dense retrieval models significantly outperform BM25, and we choose the best-performing text-embedding-ada-002 as our retreival model in our experiments.

For retrieval index construction, we observe that leveraging semantic information from both the API import paths and the first sentence of API descriptions yields the best performance. Notably, using only the first sentence of the API description outperforms using the entire description. A possible explanation is that the first sentence typically provides a concise summary of the API, which is sufficient for retrieval purposes. In contrast, the remaining content often introduces more detailed but potentially distracting information, such as parameter details and API behavior. In our

experiment, we construct retrieval index by concating the API import path and the first line of API descriptions.

We also evaluate the impact of multi-task retrieval (MT), where complex problems are decomposed into multiple subtasks, each retrieving its own relevant APIs. The retrieval results are then reranked across subtasks, and the top-k APIs are selected. Our findings indicate that MT retrieval significantly improves recall compared to single-task retrieval (ST), where the complex problem is treated as a single task to retrieve APIs.

## A.5 CONSTRUCTION OF TORCHDATA-MANUAL

The Torchdata-Manual benchmark is designed to provide complex programming problems that require the use of multiple Torchdata APIs. It follows the style of prior unseen library benchamarks (Zan et al., 2023; Ma et al., 2024b) , consisting of a natural language task description, code context, canonical solutions, and test cases. The construction process is outlined as follows:

**Torchdata API Selection.** We first curated a subset of APIs from the complete Torchdata API pool. For each problem, we randomly sampled 15 APIs from this subset, ensuring that the selected group of APIs differed from those used in previous tasks. This process helped ensure a more balanced distribution of the Torchdata APIs and maintained the variety among problems. In total, 200 groups of 15 unique APIs were selected.

**Manual Construction of Example Programming Tasks.** Two long-sequence API problems were manually written to serve as few-shot demonstration for the next step.

**LLM based Graft Generation.** We leverage GPT-4o, which has been trained on Torchdata knowledge, to craft some for programming problems for inspiration. Specifically, we provided the 2-shot demonstration and the documentation for the 15 APIs in each group, and tasked the GPT-4o with generating a programming problem that incorporated as many APIs as possible. This resulted in 200 initial problem drafts.

**Manual Curation of Programming Problems.** We manually filter out reasonable problem requirements from the drafts. Based on these filtered drafts, we then rewrote high-quality, coherent problems. In total, 50 programming problems were constructed.

**Expert Review.** Finally, we invited two Python programmers, each with four years of experience, to review the dataset and suggest adjustments. This step ensured the overall quality and correctness of the benchmark.

## A.6 ADDITIONAL IMPLEMENTATION DETAILS

Torchdata is a library that facilitate multiple data processing operations. For task planning module, we ask GPT-3.5-turbo-0125 (API-untrained model) to summarize Torchdata's purpose, key concepts, and API division logic based on Torchdata's README page[8]. The summarized results are presented in Listing 2. We also extracted few-shot API invocation planners demonstrated in Listing 3 following Ma et al. (2024b)'s approach. And both information are used for invocation task planning. Unlike the detailed functionalities for each APIs, the summarization and planners demonstrations give high-level insights into the library, facilitating better planning and reasoning for LLMs (Zheng et al., 2024). We use such summarization to represent limited domain knowledge for task planning, and no further detailed API usage information is leaked for problem solving. We also demonstrate ExploraCoder's prompts in Listing 4 - 7.

Listing 2: Condensed introduction for Torchdata.

```
Torchdata is a library of common modular data loading primitives for constructing flexible data pipelines. It
    introduces composable Iterable-style and Map-style building blocks called DataPipes, which work well
    with PyTorch's DataLoader and have functionalities for loading, parsing, caching, transforming, and
    filtering datasets.
DataPipes can be composed together into datasets and support execution in various settings and execution
    backends using DataLoader2.
The library aims to make data loading components more flexible and reusable by providing a new DataLoader2
    and modularizing features of the original DataLoader into DataPipes.
DataPipes are a renaming and repurposing of the PyTorch Dataset for composed usage, allowing for easy
    chaining of transformations to reproduce sophisticated data pipelines.
```

---

[8]https://github.com/pytorch/data/blob/v0.7.1/README.md

```
DataLoader2 is a light-weight DataLoader that decouples data-manipulation functionalities from
    torch.utils.data.DataLoader and offers additional features such as checkpointing/snapshotting and
    switching backend services for high-performant operations.
```

Listing 3: We demonstrate 2 examples for API invocation planner.

```
[task]
Read the contents of a file and verify its hash value.
[subtasks]
1. Open a file using FileOpener
2. Wrap the file object using IterableWrapper
3. Check the hash value of the file using check_hash
[task]
Fetch the first line of a text file from a given URL and print it alongside the URL.
[subtasks]
1. Instantiate an OnlineReader datapipe using an IterableWrapper that holds the URL of the text file.
2. Read lines from the OnlineReader datapipe.
3. Iterate over the datapipe and output both the URL and the first line of the text file
```

Listing 4: prompt for subtask planner.

```
I will give you a task that needs interactions with external APIs. You need to break down the task into
    several subtasks that can be implemented by invoking APIs.
{library_summary}
Examples: {fewshot_examples}
Task: {Task}
Subtasks:
```

Listing 5: prompt for CoAE.

```
We have decomposed a user requirement into multiple subtasks and tested some api-calling codes for each
    subtask.
The user has prepared some external file you will need and defines the test inputs for you:
```
{example_inputs}
{code_context}\
```\
{prior_subtasks_exploration_experience}

Now you need to learn the API usage experience from previous subtasks and implement the subsequent subtask.
<subtask>{subtask_cnt}. {subtask}</subtask>

Here are some Torchdata APIs maybe useful:
{library_api_info}

Requirements:
1. Write a playground code that imports neccessary API(s), defines your own test data as input, and calls the
    APIs to implement the subtask. Wrap the code in a ```python block```.
2. For each used API, read the API description to learn the [data formats] and [semantics] of the
    input/output object. Make sure the object is converted to the correct format and semantics before
    passing it to an API.
3. Direclty use the user-defined example inputs as your playground code inputs. Make use of the explored APIs
    from prior subtasks and predefined functions for this subtask implementation.
4. You can print anywhere to check the the data or object format. Such output will be observed after
    execution.
```

Listing 6: prompt for CoAE self-debug.

```
You were writing playground codes to explore external APIs usage for a subtask. Now you encountered an error.
    You need to debug the API usage and make the code executable.

## The buggy code:
```
{buggy_code}
```
## Error message:
{error_message}

## Relevant APIs
{api_list_str}

We omit the format requirement here.
```

Listing 7: prompt for final solution generator.

```
--------- system prompt ---------
# Context #
You are a senior Python programmer. You are assigned a task to implement an incomplete function to meet
    user's requirement. You find a new external library 'Torchdata' in <<library_documents>> that is
    helpful.
To better learn the correct usage of Torchdata's APIs, you've thought of some relevant subtasks. For each
    <<subtask>>, you have first crafted a <<playground_code>> to call APIs to implement the subtask, then
    had an <<observation>> of the code's executability, execution output, and error message.
```

```
# Objective #
Now you need to implement the user <<requirement>> by importing neccessary APIs and completing the
    <<incomplete_function>>.
# Response #
Your response should contain a complete code snippet in the following format:
```python
[YOUR IMPORT HERE]
original incomplete code snippet
[YOUR COMPLETION HERE]
```
--------- user prompt ---------
You need to complete a function to meet requirement.
<requirement>
{requirement}
</requirement>
<incomplete_function>
{cg_task_prompt}
</incomplete_function>
You have explored some API usage on various subtasks:
<explorations_experience>
{subtask_exploration_list}
</explorations_experience>
Refer to relevant APIs information:
<library_documents>
{library_api_info}
</library_documents>
Now make use of the experience and supplemented APIs to complete the function.
Note that the subtasks may not directly related to the user requirement, excessive or unnecessary API calls
    may exist. But they are to help you understand the library's APIs behavior and usage.
You have to reorganize API call sequence, add your own implementation to help transforming the data format
    between API calls.
```

## A.7 CASE STUDY

Listing 8: A failed example for naive RAG. We omit the API signature and description for simplicity

```
"""
Please complete the following function, here are some APIs maybe useful:
<API>
torchdata.datapipes.iter.ParagraphAggregator
torchdata.datapipes.map.Batcher
torchdata.datapipes.iter.Batcher
torchdata.datapipes.iter.OnDiskCacheHolder
torchdata.datapipes.iter.InBatchShuffler
torchdata.datapipes.iter.BucketBatcher
torchdata.datapipes.iter.JsonParser
torchdata.datapipes.iter.Shuffler
torchdata.datapipes.map.InMemoryCacheHolder
torchdata.datapipes.iter.FullSync
torchdata.datapipes.iter.CSVParser
torchdata.datapipes.iter.Prefetcher
torchdata.datapipes.iter.OnlineReader
torchdata.datapipes.iter.CSVDictParser
torchdata.datapipes.iter.ParquetDataFrameLoader
torchdata.datapipes.map.Shuffler
torchdata.datapipes.iter.RandomSplitter
torchdata.datapipes.map.Concater
torchdata.datapipes.iter.LineReader
torchdata.datapipes.iter.FileLister
</API>
"""
from torchdata.datapipes.iter import *
from typing import List, Tuple
import torch
def process_paragraphs_map_fn(paragraph: Tuple[str, str]) -> str:
    file_name = paragraph[0]
    paragraph_content = paragraph[1]
    return paragraph_content

def build_text_data_pipe(
        root: str="./torchdata-programming-tasks/task_82",  # Directory containing text files
        batch_size: int=16,  # Batch size
    ):
    """
    Load text data from files under a given directory, process the text data into paragraphs, create
     mini-batches of paragraphs, and shuffle the batches. The processed data should be cached and held in
     memory.
    """

Your response should contain a complete code snippet wrapped in```python```
--------- GPT-3.5-turbo output ---------
```python
# Define a function to process paragraphs
def process_paragraphs_map_fn(paragraph: Tuple[str, str]) -> str:
    file_name = paragraph[0]
    paragraph_content = paragraph[1]
    return paragraph_content

# Build the text data pipeline
```

```
def build_text_data_pipe(
        root: str="./torchdata-programming-tasks/task_82",  # Directory containing text files
        batch_size: int=16,  # Batch size
    ):
    # Create a DataPipe to list files
    file_list_dp = FileLister(root=root, masks=["*.txt"])
    # Create a DataPipe to read lines from files
    line_reader_dp = LineReader(file_list_dp)
    # Create a DataPipe to aggregate lines into paragraphs
    paragraph_aggregator_dp = ParagraphAggregator(line_reader_dp)
    # Create a DataPipe to batch paragraphs
    batcher_dp = Batcher(paragraph_aggregator_dp, batch_size=batch_size)
    # Create a DataPipe to shuffle batches
    shuffler_dp = InBatchShuffler(batcher_dp)
    # Create a DataPipe to cache and hold data in memory
    cached_dp = InMemoryCacheHolder(shuffler_dp)
    return cached_dp
--------- Analysis ---------
In this case, GPT-3.5-turbo + naive RAG successfully recalled relevant APIs to generator. However, it
    hallucinate on the usage of `FileLister` and `LineReader` APIs, since `FileLister` outputs the
    filename, while `LineReader` expects file stream as input to read each line from the stream. Such
    mistake hinders the successful generation of correct code solutions for multi-API problems.
```

Listing 9: A successful example for ExploraCoder. We show the formed exploration trace $\hat{\mathcal{E}}$ in CoAE and a successful final solution generated based on it.

```
[requirement]
Load text data from files under a given directory, process the text data into paragraphs, create mini-batches
    of paragraphs, and shuffle the batches. The processed data should be cached and held in memory.
---------- Subtask 1 on CoAE ---------
[subtask]
1. List all text files in the given directory
[exploratory code]
import torchdata
from torchdata.datapipes.iter import FileLister
root = "./torchdata-programming-tasks/task_82"
file_lister = FileLister(root=root)
for file_path in file_lister:
    print(file_path)
[observation]
Status: Executable
Output:
./torchdata-programming-tasks/task_82/file_0.txt
...omit 3 lines...
./torchdata-programming-tasks/task_82/file_4.txt
---------- Subtask 2 on CoAE ---------
[subtask]
2. Open each text file and read the text data
[exploratory code]
import torchdata
from torchdata.datapipes.iter import FileLister, IoPathFileOpener, LineReader
root = "./torchdata-programming-tasks/task_82"
file_lister = FileLister(root=root)
file_opener = IoPathFileOpener(source_datapipe=file_lister)
line_reader = LineReader(source_datapipe=file_opener)
for file_name, line in line_reader:
    print(f"File: {file_name}")
    print(f"Content: {line}")
[observation]
Status: Executable
Output:
File: ./torchdata-programming-tasks/task_82/file_0.txt
Content: This is line 0 of file 0
File: ./torchdata-programming-tasks/task_82/file_0.txt
...omit 995 lines...
File: ./torchdata-programming-tasks/task_82/file_4.txt
Content: This is line 99 of file 4
---------- Subtask 3 on CoAE ---------
[subtask]
3. Process the text data into paragraphs
[exploratory code]
import torchdata
from torchdata.datapipes.iter import FileLister, IoPathFileOpener, LineReader, ParagraphAggregator
from typing import Tuple
root = "./torchdata-programming-tasks/task_82"
file_lister = FileLister(root=root)
file_opener = IoPathFileOpener(source_datapipe=file_lister)
line_reader = LineReader(source_datapipe=file_opener)
paragraph_aggregator = ParagraphAggregator(source_datapipe=line_reader)
for file_name, paragraph in paragraph_aggregator:
    print(f"File: {file_name}")
    print(f"Paragraph: {paragraph}")
[observation]
Status: Executable
Output:
File: ./torchdata-programming-tasks/task_82/file_0.txt
Paragraph: This is line 0 of file 0
This is line 1 of file 0
...omit 500 lines...
This is line 98 of file 4
```

```
This is line 99 of file 4
---------- Subtask 4 on CoAE ---------
[subtask]
4. Create mini-batches of paragraphs using torchdata APIs
[exploratory code]
import torchdata
from torchdata.datapipes.iter import FileLister, IoPathFileOpener, LineReader, ParagraphAggregator, Batcher
from typing import Tuple
root = "./torchdata-programming-tasks/task_82"
file_lister = FileLister(root=root)
file_opener = IoPathFileOpener(source_datapipe=file_lister)
line_reader = LineReader(source_datapipe=file_opener)
paragraph_aggregator = ParagraphAggregator(source_datapipe=line_reader)
batcher = Batcher(datapipe=paragraph_aggregator, batch_size=16)
for batch in batcher:
    print(batch)
[observation]
Status: Executable
Output:
[('./torchdata-programming-tasks/task_82/file_0.tx...is line 98 of file 4\nThis is line 99 of file 4')]
---------- Subtask 5 on CoAE ---------
[subtask]
5. Shuffle the batches using shuffle API
[exploratory code]
import torchdata
from torchdata.datapipes.iter import FileLister, IoPathFileOpener, LineReader, ParagraphAggregator, Batcher,
    InBatchShuffler
root = "./torchdata-programming-tasks/task_82"
file_lister = FileLister(root=root)
file_opener = IoPathFileOpener(source_datapipe=file_lister)
line_reader = LineReader(source_datapipe=file_opener)
paragraph_aggregator = ParagraphAggregator(source_datapipe=line_reader)
batcher = Batcher(datapipe=paragraph_aggregator, batch_size=16)
shuffler = InBatchShuffler(datapipe=batcher)
for shuffled_batch in shuffler:
    print(shuffled_batch)
[observation]
Status: Executable
Output:
[('./torchdata-programming-tasks/task_82/file_4.tx...is line 98 of file 2\nThis is line 99 of file 2')]
---------- Subtask 6 on CoAE ---------
[subtask]
6. Cache and hold the processed data in memory
[exploratory code]
import torchdata
from torchdata.datapipes.iter import FileLister, IoPathFileOpener, LineReader, ParagraphAggregator, Batcher,
    InMemoryCacheHolder
from typing import Tuple
root = "./torchdata-programming-tasks/task_82"
file_lister = FileLister(root=root)
file_opener = IoPathFileOpener(source_datapipe=file_lister)
line_reader = LineReader(source_datapipe=file_opener)
paragraph_aggregator = ParagraphAggregator(source_datapipe=line_reader)
batcher = Batcher(datapipe=paragraph_aggregator, batch_size=16)
cache_holder = InMemoryCacheHolder(source_dp=batcher)
for data in cache_holder:
    print(data)
[observation]
Status: Executable
Output:
[('./torchdata-programming-tasks/task_82/file_0.tx...is line 98 of file 4\nThis is line 99 of file 4')]
</explorations_experience>
Refer to relevant APIs information:
<library_documents>
{# We omit the API information for simplicity}
</library_documents>
Now make use of the experience and supplemented APIs to complete the function.
Note that the subtasks may not directly related to the user requirement, excessive or unnecessary API calls
    may exist. But they are to help you understand the library's APIs behavior and usage. You have to
    reorganize API call sequence, add your own implementation to help transforming the data format between
    API calls.
--------- Final Solution ---------
from torchdata.datapipes.iter import *
from typing import List, Tuple
import torch
def build_text_data_pipe(
        root: str="./torchdata-programming-tasks/task_82",  # Directory containing text files
        batch_size: int=16,  # Batch size
    ):
    file_lister = FileLister(root=root)
    file_opener = IoPathFileOpener(source_datapipe=file_lister)
    line_reader = LineReader(source_datapipe=file_opener)
    paragraph_aggregator = ParagraphAggregator(source_datapipe=line_reader)
    batcher = Batcher(datapipe=paragraph_aggregator, batch_size=batch_size)
    shuffler = InBatchShuffler(datapipe=batcher)
    cache_holder = InMemoryCacheHolder(source_dp=shuffler)
    return cache_holder
--------- Analysis ---------
We can find that during CoAE, ExploraCoder incrementally reuse APIs invocations from prior subtask, and
    experiments with new API invocations and observe their behaviour. Eventually, ExploraCoder manage to
    find a successful API exploration trace, helping it to generate correct final solution.
```

Listing 10: A failed example for Self-Repair.

```
[requirement]
Load text data from files under a given directory, process the text data into paragraphs, create mini-batches
    of paragraphs, and shuffle the batches. The processed data should be cached and held in memory.
--------- initial candidate ---------
[code]
from torchdata.datapipes.iter import *
from typing import List, Tuple
import torch
def process_paragraphs_map_fn(paragraph: Tuple[str, str]) -> str:
    file_name = paragraph[0]
    paragraph_content = paragraph[1]
    return paragraph_content
def build_text_data_pipe(
        root: str="./torchdata-programming-tasks/task_82",  # Directory containing text files
        batch_size: int=16,  # Batch size
    ):
    data_pipe = FileLister(root) >> IoPathFileLister() >> FileOpener() >> LineReader()
    data_pipe = data_pipe.map(process_paragraphs_map_fn) >> Batcher(batch_size=batch_size) >>
     InMemoryCacheHolder()
    data_pipe = data_pipe >> Shuffler()
    return data_pipe
list_dp = list(build_text_data_pipe(root="./torchdata-programming-tasks/task_82", batch_size=4))
[execution result]
Status: failed
Error message:
TypeError: __init__() missing 1 required positional argument: 'root'
Error traceback at:
 - line 23 in <module>: list_dp = list(build_text_data_pipe(root="./torchdata-programming-tasks/task_82",
     batch_size=4))
 - line 18 in build_text_data_pipe:     data_pipe = FileLister(root) >> IoPathFileLister() >> FileOpener() >>
     LineReader()
--------- round 1 self-repair ---------
[feedback]
The error message indicates that the `FileLister` class in the `build_text_data_pipe` function is missing the
     required positional argument `root`. The `FileLister` class expects the root directory containing text
     files as an argument. To resolve this error, make sure to provide the `root` argument when creating the
     `FileLister` instance in the `build_text_data_pipe` function.
[code]
from torchdata.datapipes.iter import *
from typing import List, Tuple
import torch

def process_paragraphs_map_fn(paragraph: Tuple[str, str]) -> str:
    file_name = paragraph[0]
    paragraph_content = paragraph[1]
    return paragraph_content

def build_text_data_pipe(
        root: str="./torchdata-programming-tasks/task_82",  # Directory containing text files
        batch_size: int=16,  # Batch size
    ):
    """
    Load text data from files under a given directory, process the text data into paragraphs, create
     mini-batches of paragraphs, and shuffle the batches. The processed data should be cached and held in
     memory.
    """
    data_pipe = FileLister(root=root) >> IoPathFileLister() >> FileOpener() >> LineReader()
    data_pipe = data_pipe.map(process_paragraphs_map_fn) >> Batcher(batch_size=batch_size) >>
     InMemoryCacheHolder()
    data_pipe = data_pipe >> Shuffler()

    return data_pipe
list_dp = list(build_text_data_pipe(root="./torchdata-programming-tasks/task_82", batch_size=4))
Status: failed
Error message:
TypeError: __init__() missing 1 required positional argument: 'root'
Error traceback at:
 - line 23 in <module>: list_dp = list(build_text_data_pipe(root="./torchdata-programming-tasks/task_82",
     batch_size=4))
 - line 18 in build_text_data_pipe:     data_pipe = FileLister(root=root) >> IoPathFileLister() >>
     FileOpener() >> LineReader()
--------- round 2 self-repair ---------
[feedback]
The error message indicates that the `FileLister` class in the `build_text_data_pipe` function is missing the
     required positional argument `root`. The `FileLister` class expects the root directory containing text
     files as an argument. To resolve this error, make sure to provide the `root` argument when creating the
     `FileLister` instance in the `build_text_data_pipe` function.
[code]
from torchdata.datapipes.iter import *
from typing import List, Tuple
import torch
def process_paragraphs_map_fn(paragraph: Tuple[str, str]) -> str:
    file_name = paragraph[0]
    paragraph_content = paragraph[1]
    return paragraph_content
def build_text_data_pipe(
        root: str="./torchdata-programming-tasks/task_82",  # Directory containing text files
        batch_size: int=16,  # Batch size
    ):
    """
```

```
1188
1189        Load text data from files under a given directory, process the text data into paragraphs, create
1190        mini-batches of paragraphs, and shuffle the batches. The processed data should be cached and held in
1191        memory.
           """
1192        data_pipe = FileLister(root=root) >> IoPathFileLister() >> FileOpener() >> LineReader()
           data_pipe = data_pipe.map(process_paragraphs_map_fn) >> Batcher(batch_size=batch_size) >>
1193        InMemoryCacheHolder()
1194        data_pipe = data_pipe >> Shuffler()
           return data_pipe
1195    list_dp = list(build_text_data_pipe(root="./torchdata-programming-tasks/task_82", batch_size=4))
       [execution result]
1196    Status: failed
       Error message:
1197    TypeError: __init__() missing 1 required positional argument: 'root'
       Error traceback at:
1198     - line 23 in <module>: list_dp = list(build_text_data_pipe(root="./torchdata-programming-tasks/task_82",
           batch_size=4))
1199     - line 18 in build_text_data_pipe:    data_pipe = FileLister(root=root) >> IoPathFileLister() >>
           FileOpener() >> LineReader()
1200    --------- round 3 self-repair ---------
       [feedback]
1201    The error message indicates that the 'build_text_data_pipe' function is missing the required positional
           argument 'root'. In this case, the 'FileLister' class expects the 'root' argument to be provided. To
1202        resolve this issue, make sure to pass the 'root' argument when calling the 'build_text_data_pipe'
           function.
1203    [code]
       from torchdata.datapipes.iter import *
1204    from typing import List, Tuple
       import torch
1205    def process_paragraphs_map_fn(paragraph: Tuple[str, str]) -> str:
1206        file_name = paragraph[0]
           paragraph_content = paragraph[1]
1207        return paragraph_content
       def build_text_data_pipe(
1208            root: str = "./torchdata-programming-tasks/task_82",  # Directory containing text files
1209            batch_size: int = 16,  # Batch size
       ):
1210        """
1211        Load text data from files under a given directory, process the text data into paragraphs, create
           mini-batches of paragraphs, and shuffle the batches. The processed data should be cached and held in
1212        memory.
           """
1213        data_pipe = FileLister(root=root) >> IoPathFileLister() >> FileOpener() >> LineReader()
           data_pipe = data_pipe.map(process_paragraphs_map_fn) >> Batcher(batch_size=batch_size) >>
1214        InMemoryCacheHolder()
1215        data_pipe = data_pipe >> Shuffler()
           return data_pipe
1216    list_dp = list(build_text_data_pipe(root="./torchdata-programming-tasks/task_82", batch_size=4))
       --------- round 4-5 self-repair ---------
1217    ...omit for simplicity...
       --------- round 6 self-repair ---------
1218    [feedback]
1219    The error message indicates that the 'root' argument is missing when creating the FileLister data pipe. The
           FileLister API expects the root argument to be provided, but it is not being passed correctly in the
1220        build_text_data_pipe function. To resolve this issue, ensure that the 'root' argument is correctly
           passed when creating the FileLister data pipe within the build_text_data_pipe function.
1221    [code]
       from torchdata.datapipes.iter import *
1222    from typing import List, Tuple
       import torch
1223
1224    def process_paragraphs_map_fn(paragraph: Tuple[str, str]) -> str:
1225        file_name = paragraph[0]
           paragraph_content = paragraph[1]
1226        return paragraph_content
1227    def build_text_data_pipe(
            root: str = "./torchdata-programming-tasks/task_82",  # Directory containing text files
1228        batch_size: int = 16,  # Batch size
       ):
1229        """
1230        Load text data from files under a given directory, process the text data into paragraphs, create
           mini-batches of paragraphs, and shuffle the batches. The processed data should be cached and held in
1231        memory.
           """
1232        data_pipe = FileLister(root=[root]) >> IoPathFileLister() >> FileOpener() >> LineReader()
           data_pipe = data_pipe.map(process_paragraphs_map_fn) >> Batcher(batch_size=batch_size) >>
1233        InMemoryCacheHolder()
1234        data_pipe = data_pipe >> Shuffler()
           return data_pipe
1235    list_dp = list(build_text_data_pipe(root="./torchdata-programming-tasks/task_82", batch_size=4))
1236    [execution result]
       Status: failed
1237    Error message:
       TypeError: __init__() missing 1 required positional argument: 'root'
1238    Error traceback at:
        - line 23 in <module>: list_dp = list(build_text_data_pipe(root="./torchdata-programming-tasks/task_82",
1239        batch_size=4))
        - line 18 in build_text_data_pipe:    data_pipe = FileLister(root=[root]) >> IoPathFileLister() >>
1240        FileOpener() >> LineReader()
1241    --------- Analysis ---------
```

```
In the initial code solution, Self-Repair severely misuses the Torchdata library. Specifically, it
    incorrectly assumes that Torchdata APIs can be chained using \">>, and this incorrect API usage pattern
    propagates throughout the entire LLM-decoded API invocation sequence. Despite receiving clear error
    messages in each round of Self-Repair, the system fails to correct the code because its initial
    solution deviates significantly from the canonical implementation.
```

Listing 11: A successful example for self-debug on ExploraCoder.

```
[API invocation subtask]
Load the text data from the zip file using load_text_data_from_zip
--------- failed experimental code ---------
[code]
import torchdata
from torchdata.datapipes.iter import FileLister, ZipArchiveLoader, FileOpener
from torchdata.datapipes.utils.common import StreamWrapper

# define test inputs
root = './torchdata-programming-tasks/task_89'
file_path = './torchdata-programming-tasks/task_89/file_0.zip'

# Load text data from the zip file
file_stream = open(file_path, 'rb')
zip_loader = ZipArchiveLoader([(file_path, StreamWrapper(file_stream))])
file_opener = FileOpener(zip_loader)

# Iterate over the file opener to get text data
for file_info in file_opener:
    file_path, file_stream = file_info
    text_data = file_stream.read()
    print(f"Text Data from {file_path}: {text_data}")

file_stream.close()
[onservation]
Status: failed
Output:
Error Message:
TypeError: Expected string type for pathname, but got <class 'tuple'>
This exception is thrown by __iter__ of FileOpenerIterDataPipe(datapipe=ZipArchiveLoaderIterDataPipe,
    encoding=None, length=-1, mode='r')
Error traceback at:
 - line 15 in <module>: for file_info in file_opener:
 - line 212 in get_file_binaries_from_pathnames: raise TypeError("Expected string type for pathname, but got
    {}"

--------- fixed by self-debug ---------
[code]
import torchdata
from torchdata.datapipes.iter import FileLister, ZipArchiveLoader, FileOpener
from torchdata.datapipes.utils.common import StreamWrapper

# User-defined test inputs
root = './torchdata-programming-tasks/task_89'
file_path = './torchdata-programming-tasks/task_89/file_0.zip'

# Load text data from the zip file
file_stream = open(file_path, 'rb')
zip_loader = ZipArchiveLoader([(file_path, StreamWrapper(file_stream))])

# Iterate over the zip loader to get text data
for file_info in zip_loader:
    file_path, file_stream = file_info
    text_data = file_stream.read()
    print(f"Text Data from {file_path}: {text_data}")
[observation]
Status: success
Output:
Text Data from torchdata-programming-tasks/task_89/file_0.zip/file_0_0.txt: b'...'
...omit 3 lines...
tasks/task_89/file_0.zip/file_0_4.txt: b'...'
--------- Analysis ---------
In ExploraCoder's initial experimental code, it incorrectly hallucinates the usage of FileOpener. However,
    after a round of self-debugging, ExploraCoder is able to correct this simple API misuse and
    successfully observe behavior from the correct API invocation.
```

## A.8 THE EFFECTIVENESS OF OUR TASK PLANNING

Although It is hard to directly quantify the quality of decomposed tasks' granularity, we can evaluate it indirectly by calculating the number of APIs included in each subtask, since our design aims to ensure each decomposed subtask involves 1-2 API explorations, so that it's easy enough to be solved.

As shown in Table 10, the average number of decomposed subtasks by GPT-3.5 is closely aligned with the average number of API invloved across two datasets. This indicates that the decomposition strategy effectively achieves the desired granularity. The ExploraCoder's overall performance also indicates the effectiveness of our task planning.

|                   | #API invocation | #decomposed subtask | API per subtask |
|-------------------|-----------------|---------------------|-----------------|
| Torchdata-Github  | 4.26            | 4.06                | 1.04            |
| Torchdata-Manual  | 9.94            | 8.22                | 1.21            |

Table 10: Summary of decomposed subtask statistics.

Table 11: Comparing ExploraCoder with baseline approaches using GPT3.5 on Torchdata-Github.

| Method              | $k = 1$ |         | $k = 5$ |         | $k = 10$ |         | $k = 20$ |         |
|---------------------|---------|---------|---------|---------|----------|---------|----------|---------|
|                     | Pass    | Success | Pass    | Success | Pass     | Success | Pass     | Success |
| Direct              | 0%      | 0%      | 0%      | 0%      | 0%       | 0%      | 0%       | 0%      |
| Naive RAG           | 0.28%   | 0.75%   | 1.31%   | 3.64%   | 2.38%    | 7.05%   | 3.72%    | 13.16%  |
| CAPIR               | 3.26%   | 5.78%   | 8.33%   | 10.81%  | 10.64%   | 17.41%  | 13.33%   | 26.70%  |
| ExploraCoder        | 9.43%   | 19.89%  | 14.38%  | 27.17%  | 16.07%   | 28.35%  | 17.55%   | 29.32%  |
| CAPIR + Self-Repair | 9.61%   | 16.52%  | 9.80%   | 19.87%  | 9.80%    | 21.26%  | 9.80%    | 23.32%  |
| ExploraCoder*       | 14.24%  | 27.22%  | 22.60%  | 40.21%  | 24.99%   | 43.71%  | 27.22%   | 46.51%  |

## A.9 BASELINE COMPARISON ON EXTENDED TORCHDATA-MANUAL

We extended the dataset of Torchdata-Manual and we conduct the baseline experiments on the extended samples. The results are povided in Table 11. The trends observed in the results are consistent with the main experiment, with ExploraCoder achieving SOTA performance. This consistency demonstrates the robustness of the proposed ExploraCoder and its evaluation.

## A.10 DIFFERENCE FROM SWE-BENCH CODE AGENTS

In this section, we address the differences between our framework, ExploraCoder, and the recently popular Swe-Bench code agents. While there may be some similarities in certain elements, we highlight key distinctions in application focus and methodology that clearly differentiate our work.

**Shared Elements.** ExploraCoder shares certain widely adopted concepts and paradigms (such as planning, Chain-of-Thought (CoT) reasoning, Retrieval-Augmented Generation (RAG), and execution) with Swe-Agent (Yang et al., 2024), AutoCodeRover (Zhang et al., 2024), and other advanced Swe-Bench agents (Liu et al., 2024b; Ma et al., 2024a). However, these paradigms are commonly used in recent research (Le et al., 2024; Ni et al., 2024) and were not originally proposed by these works. Thus, overlapping adoption of these widely accepted elements does not suggest significant resemblance between our framework and theirs. The novelty and improvements in ExploraCoder's element designs are further discussed in Appendix A.11.

**Scenario Differences.** The application focus of ExploraCoder is fundamentally different from that of Swe-Bench agents. Swe-Bench agents primarily rely on a small, predefined set of repository tools to identify buggy lines and edit existing code. In contrast, ExploraCoder synthesizes complex, unseen API invocation code. Using the terminology of agentic systems, our framework addresses the challenge of selecting appropriate tools from a large, previously unseen toolset and ensuring their correct usage—a scenario that falls outside the scope of Swe-Bench agents.

**Methodological Differences.** ExploraCoder employs a fundamentally different methodology, taking a non-agent approach. According to the definition in Agentless (Xia et al., 2024), agent systems follow a "reason-then-act" style, iteratively planning and acting based on common-sense knowledge. ExploraCoder, however, utilizes a document-enhanced planner specifically tailored for domain-specific planning logic. This planner generates an integrated set of library-aware subtasks in the absence of full API list, as detailed in Section 3.2. This distinct design sets ExploraCoder apart in terms of both framework pipeline and methodology.

## A.11 SUMMARIZATION OF CONTRIBUTION

**(1) A novel step-wise generation method**: We designed a novel Chain-of-API-Exploration (CoAE) mechanism for step-wise code generation. In each intermediate step, CoAE generates API invo-

cations based on limited API documentation and leverages executability signals to rectify coding decisions in real time.

The key difference between CoAE and other (CoT-inspired and execution-driven) iterative code generation methods, such as CodeChain (Le et al., 2024), Self-Debug (Chen et al., 2023), and Swe-Agent (Yang et al., 2024), lies in that it focuses on each intermediate subtask within a targeted integral code solution and uses execution as step-wise reward signal. Specifically, Self-Debug and Swe-Agent applies end-to-end code/patch generation and iteratively edit the code/patch based on new information from executing the code/reproduction script. CodeChain generates modularized code snippets but does not consider step-wise execution signals and fails to effectively utilize the logical relationships between modules.

**(2) An Exploratory-Programming-inspired Framework**: We designed a unified framework, ExploraCoder, to integrate CoAE with other effective techniques. ExploraCoder is inspired by the exploratory programming paradigm observed in human developers (Beth Kery & Myers, 2017), who actively read API documents and gather more API usage experience from trial execution.

While ExploraCoder integrates existing techniques such as CoT reasoning, RAG, and execution-then-debug, its framework design introduces **non-trivial innovations** beyond merely applying these techniques to the domain of unseen API usage. Specifically:

- **Integration of CoT reasoning**: Unlike the trivial decomposition of subtasks as CoT instructions, ExploraCoder enhances CoT reasoning by utilizing the exploration trace produced by CoAE as enriched CoT instructions.

- **Debugging mechanisms**: ExploraCoder incorporates debugging at each CoAE intermediate step, rather than debugging the final solution as done in CAPIR+Self-Repair or many Swe-Bench agents.

- **API retrieval**: Beyond retrieving API docs based solely on semantic similarity, we refine the recommended API set by dynamically parsing the CoAE steps to identify APIs that are more relevant and usable.

Experimental results also indicates the superiority of ExploraCoder's design choice compared to the aforementioned trivial technique application.

**(3) A new multi-API-invocation benchmark**: We manually constructed a new benchmark, Torchdata-Manual, containing 100 complex programming problems each involving 8-14 Torchdata API invocations. To the best of our knowledge, Torchdata-Manual contains the longest API sequences among publicly reported library-oriented benchmarks.

