# OpenReview forum: "ExploraCoder: Advancing code generation for multiple unseen APIs via planning and chained exploration"
_ICLR.cc/2025/Conference — ICLR 2025 Conference Withdrawn Submission_

### Official Review · Reviewer_XmiU · 2024-11-02

**Soundness:** 2
**Presentation:** 2
**Contribution:** 1
**Rating:** 3
**Confidence:** 5

**Summary:**

This paper introduces ExploraCoder, a framework that enables Large Language Models (LLMs) to generate code involving multiple unseen APIs through planning and exploration. The framework breaks down complex programming tasks into simpler subtasks, explores API usage through experimental code generation and execution, and leverages successful experiences to guide final solution generation. The authors curated two benchmarks: Torchdata-Github and the more chanllenging Torchdata-Manual. Authors evaluate their approach on two benchmarks, demonstrating significant improvements over baseline methods and conducting comprehensive ablation studies to validate their design choices.

**Strengths:**

- The problem was formulated clearly as a practical challenge in code generation
- The authors explain their proposed method clearly
- Experiments show improvement over baseline models that the authors opt to compare against
- Various ablations especially on how different components contribute to the system's performance

**Weaknesses:**

The most significant concern I have is the limited novelty of this paper. The proposed ExploraCoder framework seems closely resembling existing code agent frameworks that can perform operations that cover what has been discussed in the paper, e.g., planning, reading code and code documentation, chain/tree-of-thought reasoing, and code execution, and iterations of all above. The core components of the proposed method, e.g., task planning, API recommendation, Chain of API Exploration, solution generator, etc. are methodologically similar to existing approaches in code agents. The paper does not sufficiently demonstrate how its approach fundamentally differs from or improves upon these existing methods, beyond applying similar techniques to the specific domain of unseen API usage.

A related weakness is the paper's related work. The authors have notably omitted discussion of recent advances in code agents, e.g., from swe-agent to AutoCodeRover to the leading ones on swebench. Without comparing against these relevant approaches, it's difficult to assess the true contribution of ExploraCoder to the field.

Further, the comparison in experiments setting might not be fair. the proposed method requires significantly more model calls than the baseline approaches noted in the paper, as it involves multiple rounds of code generation, execution, and debugging for each subtask. However, this aspect is neither analyzed nor discussed in the paper. The authors should provide a detailed and transparent analysis of the computational overhead.

Finally, both benchmarks mentioned in the paper (Torchdata-Github and Torchdata-Manual) contain only 50 problems each, which is relatively small for drawing statistically significant conclusions. While the problems themselves may be complex and requires multiple API usage, the robustness and generalizability of the results are debatable given the small size of the benchmark and domain.

**Questions:**

See weaknesses above for the most critical ones on novelty, comparison of baseline methods, fairness in benchmarking, and scalability of the dataset.

In addition,

- Could the authors provide qualititive examples on 1) the benchmarks and 2) where the proposed method demonstrate improvements over baselines?

- On the dataset curation part (Line 843 onwards in A.5): could you elaboreate on 1) how the example programming tasks was created, and 2) how the expert review was done?

- A lot of useful content has been presented in appendix only. This creates troubles in the flow of the presentation. Please consider re-arrange the paper to make sure that the most critical and logically important topics are presented in the main content without the need of frequently referring to the Appendix.

- nit: there are some typos throughout the paper. Please fix them in next iteration. Some examples: 1) L30 niave->naive 2) L116 the citation should be We et al. 2022 instead of 2024. 3) L397: abosolue->absolute 4) L524 performe->perform 5) L922 sematics->sematics

---

> ### Author Response · Authors · 2024-11-24
> **Response Summarization**
>
> Dear reviewer XmiU, we deeply appreciate your acknowledgment and support of our work. Below, we respond to all the raised weaknesses (W) and questions (Q) with the following organizations:
>
> 1. W1 & W2: Novelty and Contribution
>     1. Difference from Swe-bench Code Agents
>     2. Summarization of Contribution
> 2. W3: Analysis of Computational Overhead and Fairness of Baseline
> 3. W4: Scalability of the Dataset
>     1. Small Size Benchmark Can Provide Statistically Significant Conclusion
>     2. The Choice of Torchdata Domain
>     3. The Benchmark is now Extended
> 4. Q1: Qualitative Examples of Benchmark and Methods.
> 5. Q2: Elaboration on the Dataset Construction Details
> 6. Q3 & Q4: Typos and Content Organization

---

> ### Author Response · Authors · 2024-11-24
> **(Part 1/2 of) W1 & W2：Novelty and Contribution:**
>
> ## Difference from Swe-bench code agents
>
> We would like to address your concern regarding the perceived resemblance of our framework to existing swebench code agents, and clarify why these works were not discussed in the related work section.
>
> 1. **A certain of overlapped element-adoption design does not mean a significant resemblance.** ExploraCoder shares certain elements (planning, CoT, RAG, execution) with Swe-Agent, AutoCodeRover, (and many other SweBench agents[1][2]). However, these elements are widely applied concept/paradigms in a series of recent research [3][4], and are not originally proposed by code agents. We think it is inappropriate to judge our work as resembling Code Agent solely based on a few overlapping element-adoption design. We discuss our novelty and improvements of our components in the “Summarization of contribution” later.
> 2. **Scenical difference of ExploraCoder and Swebench agents.** Moreover, SweBench agents essentially leverages a small, predefined set of repository tools to identify buggy lines and edit existing code. **This focus is fundamentally different** from our work, which aims to **synthesize complex unseen API invocation code.** In the context of agentic systems, our study could be roughly characterized as exploring how to select the appropriate tools from a large unseen toolset and make correct tool use. (Though ExploraCoder is not an agentic system)
> 3. **Methodological difference of ExploraCoder and Swebench agents.** ExploraCoder is a non-agent approach (Following Agentless[5]’s definition, agent systems apply “reason then act” style iterations, where agent use common-sense knowledge to plan one step at a time). Instead, we employ an document-enhanced planner specifically tailored for domain-specific planning, which plans an integral set of library-aware subtasks in the absence of full API list (usually too large to put in context), as described in Section 3.2 (lines 180–190) of the original manuscript. This makes **our ExploraCoder essentially different from code agents in terms of framework pipeline.**
>
> Overall, we think our ExploraCoder is scenically and methodologically different from Swe-bench code agents. Previously, we did not include Swe-bench agents in our related work section for secondary relevancy. Following your suggestion, **we now have updated the discussion of comparing ExploraCoder with Swe-bench code agents in Appendix A.10** in the revised manuscript, and cited relevant work.
>
> [1] MarsCode Agent: AI-native Automated Bug Fixing. Liu et al. https://arxiv.org/abs/2409.00899
>
> [2] How to Understand Whole Software Repository?. Ma et al. https://arxiv.org/abs/2406.01422
>
> [3] CodeChain: Towards Modular Code Generation Through Chain of Self-revisions with Representative Sub-modules, Le et al. https://arxiv.org/abs/2310.08992
>
> [4] NExT: Teaching Large Language Models to Reason about Code Execution. Ni et al. https://arxiv.org/abs/2404.14662
>
> [5] Agentless: Demystifying LLM-based Software Engineering Agents. Xia et al. https://arxiv.org/abs/2407.01489

---

> ### Author Response · Authors · 2024-11-24
> **(Part 2/2 of) W1 & W2：Novelty and Contribution:**
>
> ## Summarization of contribution
>
> We sincerely thank you for highlighting the lack of clarity regarding our contributions, as mentioned in “*The paper does not sufficiently demonstrate how its approach fundamentally differs from or improves upon these existing methods … it’s difficult to assess the true contribution of ExploraCoder to the field.*”
>
> We’d like to reorganize our contributions here:
>
> **(1) A novel step-wise generation method**：We designed a novel **Chain-of-API-Exploration (CoAE)** mechanism for step-wise code generation. In each intermediate step, CoAE generates API invocations based on limited API documentation and leverages executability signals to rectify coding decisions in real time.
>
> The key difference between CoAE and other (CoT-inspired and execution-driven) iterative code generation methods, such as CodeChain[1], Self-Debug[2], and Swe-Agent[3], lies in that it **focuses on each intermediate subtask within a targeted integral code solution** and uses execution as step-wise reward signal. Specifically, Self-Debug and Swe-Agent applies end-to-end code/patch generation and iteratively edit the code/patch based on new information from executing the code/reproduction script. CodeChain generates modularized code snippets but does not consider step-wise execution signals and fails to effectively utilize the logical relationships between modules.
>
> **(2) An Exploratory-Programming-inspired Framework**: We designed a unified framework, **ExploraCoder**, to integrate CoAE with other effective techniques. ExploraCoder is inspired by the **exploratory programming paradigm** observed in human developers[4], who actively read API documents and gather more API usage experience from trial execution.
>
> In response to your comment that “*beyond applying similar techniques to the specific domain of unseen API usage.”*, While ExploraCoder integrates existing techniques such as CoT reasoning, RAG, and execution-then-debug, its framework design introduces **non-trivial innovations** beyond merely applying these techniques to the domain of unseen API usage. Specifically:
>
> - **Integration of CoT reasoning:** Unlike the trivial decomposition of subtasks as CoT instructions (e.g., EpiGen), ExploraCoder enhances CoT reasoning by utilizing the **exploration trace** produced by CoAE as enriched CoT instructions.
> - **Debugging mechanisms:** ExploraCoder incorporates debugging at each CoAE intermediate step, rather than debugging the final solution as done in CAPIR+Self-Repair (a baseline in our original manuscript) or many SweBench agents.
> - **API retrieval:** Beyond retrieving API docs based solely on semantic similarity (as done in most previous work), we refine the recommended API set by dynamically parsing the CoAE steps to identify APIs that are more relevant and usable.
>
> Experimental results also indicates **the superiority of ExploraCoder’s design choice compared to the aforementioned trivial technique application.**
>
> **(3) A new multi-API-invocation benchmark**：We manually constructed a new benchmark, **Torchdata-Manual**, containing 100 (50 in our original manuscript) complex programming problems each involving 8-14 Torchdata API invocations. To our best knowledge, **Torchdata-Manual contains the longest API sequences among publicly available library-oriented benchmarks.**
>
> The detailed summarization of our contribution have been updated in our manuscript **Appendix A.11**
>
> ---
>
> [1] CodeChain: Towards Modular Code Generation Through Chain of Self-revisions with Representative Sub-modules, Le et al. https://arxiv.org/abs/2310.08992
>
> [2] Teaching Large Language Models to Self-Debug. Chen et al. https://arxiv.org/abs/2304.05128
>
> [3] SWE-agent: Agent-Computer Interfaces Enable Automated Software Engineering. Yang et al. https://arxiv.org/abs/2405.15793
>
> [4] Exploring Exploratory Programming. Kery et al. https://api.semanticscholar.org/CorpusID:21574188

---

> ### Author Response · Authors · 2024-11-24
> **W3: Analysis of Computational Overhead and Fairness of Baseline**
>
> ## Fairness of benchmarking debug mechanisms
>
> We agree that computational overhead is a critical factor, in fact, we have strived to provide a fair comparison in the original manuscript. As we stated in Section 5.4, we dynamically aligns the budgets of two debug mechanisms at each problem.
>
> > We compare ExploraCoder and Self-Repair under the same computational budget. Specifically, for each problem, ExploraCoder generates n plans, enabling up to n debug operations in CoAE, we set the iteration budget for Self-repair to n accordingly.
> >
>
> ## Analysis of the computational overhead and comparison in terms of model call
>
> In response to your comment, “*the proposed method requires significantly more model calls than the baseline approaches noted in the paper… The authors should provide a detailed and transparent analysis of the computational overhead.*“, we present a detailed breakdown of the computational overhead (model call) and corresponding performance for our method and the baselines in tables below (calculation details are provided at the end of our reply). We can observe:
>
> 1. Iterative code generation approaches(last 3 rows) involves more model call compared to single-pass generation(first 3 rows) but achieves higher performance, which is an intuitive trade-off between computational cost and model efficacy.
> 2. **Performance improvements stem more from** **methodological design.** higher model call does not necessarily lead to better performance. For example, while CAPIR+Self-Repair requires more model calls than ExploraCoder, our method achieves superior performance. This highlights that performance improvements stem more from methodological design rather than simply increasing computational resources.
>
> We then **perform a stricter comparison under similar model call conditions** (we sincerely appreciate your suggestion), we analyzed the three iterative-style generation approaches in details. **ExploraCoder achieves superior performance compared to CAPIR+Self-Repair, even consuming fewer models calls and without incorporating a debug mechanism**；ExploraCoder* achieves more significant performance gains through additional model calls.
>
> |  | model call(pre-processing) |  model call(code generation) | model call(total) | pass@10 |
> | --- | --- | --- | --- | --- |
> | naiveRAG | 0 | 1 | 1 | 7.81 |
> | EpiGen | n+2 | 1 | 3+n | 10.49 |
> | CAPIR | n+2 | 1 | 3+n | 12.19 |
> | ExploraCoder | n+2 | n+1 | 3+2n | 16.64 |
> | CAPIR + self-repair | n+2 | 1.5n+2 | 4+2.5n | 14.62 |
> | ExploraCoder* | n+2 | 2.6n+1 | 3+3.6n | 22.19 |
>
> ---
>
> ***calculation details of the table**
>
> For model calls, we provide clear analytical expressions based on $n$, the number of decomposed subtasks. For the two approaches involving self-debug mechanisms, Self-Repair and ExploraCoder*, number of debugging model calls are not deterministic. Therefore we use the formulated expectation based on the empirically observed debug rate in our experiments. Let’s set the probability of the two methods conducting debug as $p_1$ (ExploraCoder*) and $p_2$ (Self-repair). Their expectation of model call can be formulated as $(1+5p_1)n+1$ and $2p_2n+2$. Notably, $p_2$=0.75 is significantly higher than $p_1$=0.32 across two benchmark. This debug rate difference arises because ExploraCoder* focuses on debugging simple subtasks (which are generally less error-prone), while Self-Repair always attempts to debug a complete solutions (which often fail to repair successfully, triggering additional debugging iterations up to the budget limit).

---

> ### Author Response · Authors · 2024-11-24
> **(Part 1/2 of) W4: Scalability of the Dataset**
>
> ## Small size benchmark can provide statistically significant conclusion
>
> We would like to address the concern raised in: “both benchmarks mentioned in the paper (Torchdata-Github and Torchdata-Manual) contain only 50 problems each, which is relatively small for drawing statistically significant conclusions.”
>
> Although the dataset size may seem relatively small, it is important to emphasize that many widely recognized benchmarks in the field of LLM-based code generation are similarly scaled and consist of manually curated datasets, which have proven valuable for evaluation. Examples include **HumanEval** with 164 samples [1], **CodeContests** with 165 samples [2], **Torchdata-Eval** with 50 samples [3], and **PandasEval** and **NumpyEval**, each containing 101 samples [4].
>
> As discussed in [1][2], **the robustness of benchmarking on small-scale datasets is ensured by the use of normalized pass rate evaluation metrics, pass@k,** where multiple candidates (n > k) are sampled to calculate the expected success rate within k trials. Thus, **we believe our 100 carefully curated, high-quality problems provide meaningful and statistically significant results** for evaluating LLM-based code generation.
>
> ## The choice of Torchdata domain
>
> We would also like to clarify the choice of our Torchdata-based benchmark.
>
> **1. Torchdata-based evaluation aligns with prior work.** Firstly, we would like to clarify the choice of our Torchdata-based benchmark. Our selection aligns with prior work on unseen-library tasks [3][5][6][7] and does not imply a specific preference for the Torchdata library. Torchdata was selected due to its unique combination of qualities, which are currently **difficult to identify in other open-source libraries**:
>
> - **Limited Pretraining Exposure**, making it suitable for conducting evaluation on unseen API invocation tasks.
>
> - **Rich and Diverse APIs**, making it suitable for constructing a multi-API benchmark with diverse and meaningful evaluations.
>
> - **Well-Maintained Documentation,** making it particularly suitable for retrieval-based code generation methods.
>
> **2. Representativeness of Torchdata as a Benchmark.** Secondly, Torchdata holds a certain level of representativeness among external libraries. Torchdata provides practical data processing APIs for Pytorch, which is objectively and functionally similar to many other commonly used libraries such as Pandas, Numpy, Huggingface Datasets, etc. (Unfortunately, their API knowledge has already been exposed to advanced LLMs), and could resemble many privately-maintained or up-coming libraries. Therefore, we believe that **evaluating ExploraCoder on Torchdata serves as a meaningful reference for its applicability to other libraries.**
>
> Moreover, to enhance the **diversity and generalizability** of our benchmark, we also sampled a variety of API invocations and combinations from the Torchdata API pool as mentioned in our original manuscript.
>
> **3. ExploraCoder is Torchdata-Agnostic.** Additionally, our proposed ExploraCoder is general plug-and-play framework and does not rely on any Torchdata-specific steps. As such, it **can be applied to other library’s programming problems.**
>
> [1] Evaluating Large Language Models Trained on Code. Chen et al. https://arxiv.org/abs/2107.03374
>
> [2] Competition-Level Code Generation with AlphaCode. Li et al. https://arxiv.org/abs/2203.07814
>
> [3] When Language Model Meets Private Library. Zan et al. https://arxiv.org/abs/2210.17236
>
> [4] CERT: Continual Pre-Training on Sketches for Library-Oriented Code Generation. Zan et al. http://arxiv.org/abs/2206.06888
>
> [5] Private-Library-Oriented Code Generation with Large Language Models. Zan et al. https://arxiv.org/abs/2307.15370
>
> [6] Compositional API Recommendation for Library-Oriented Code Generation. Ma et al. https://arxiv.org/abs/2402.19431
>
> [7] ToolCoder: Teach Code Generation Models to use API search tools. Zhang et al. https://arxiv.org/abs/2305.04032

---

> ### Author Response · Authors · 2024-11-24
> **(Part 2/2 of) W4: Scalability of the dataset**
>
> ## The benchmark is now extended
>
> To further address the reviewers’ concerns regarding the size of our benchmark, we have expanded the Torchdata-Manual dataset. Using the same methodology as before, we manually curated an additional 50 complex programming problems that are distinct from the original dataset. As a result, Torchdata-Manual now contains 100 problems. Combined with the 50 problems in Torchdata-Github, **our Torchdata series benchmark now includes a total of 150 problems.**. The remaining experiments in the original manuscript will be conducted on the extended dataset, and the results will be updated into the manuscripts.
>
> We have conducted experiments on representative baselines using GPT-3.5-turbo (the primary base model used in our original paper), and the results are presented below:
>
> |  | pass@1 | success@1 | pass@5 | success@5 | pass@10 | success@10 | pass@20 | success@20 |
> | --- | --- | --- | --- | --- | --- | --- | --- | --- |
> | direct | 0 | 0 | 0 | 0 | 0 | 0 | 0 | 0 |
> | naive RAG | 0.28 | 0.75 | 1.31 | 3.64 | 2.38 | 7.05 | 3.72 | 13.16 |
> | CAPIR | 3.26 | 5.78 | 8.33 | 10.81 | 10.64 | 17.41 | 13.33 | 26.70 |
> | ExploraCoder | 9.43 | 19.89 | 14.38 | 27.17 | 16.07 | 28.35 | 17.55 | 29.32 |
> | CAPIR+self-repair | 9.61 | 16.52 | 9.80 | 19.87 | 9.80 | 21.26 | 9.80 | 23.32 |
> | ExploraCoder* | 14.24 | 27.22 | 22.60 | 40.21 | 24.99 | 43.71 | 27.22 | 46.51 |
>
> The trends observed in the results are consistent with the original paper, with ExploraCoder achieving SOTA performance. This **consistency** allows us to draw similar analytical conclusions, **demonstrating the robustness of our dataset.** This result is updated in **Appendix A.9.**. The remaining experiments in the original manuscript will be conducted on the extended dataset, and the results will be updated into the manuscripts.

---

> ### Author Response · Authors · 2024-11-24
> **Q1: Qualitative Examples of Benchmark and Methods.**
>
> We have provided a series of case studies in our original manuscript’s Appendix A.7, from Listing 8 to Listing 10. This includes different methods (naive RAG, ExploraCoder, Self-Repair) solving the same example problem from our benchmark. We also provided an case study of ExploraCoder* in Listing 10, where we demonstrate the self-debug trace at each intermediate subtask in CoAE.
>
> We put the main analysis results here:
>
> In summary, ExploraCoder outperforms the baseline by employing step-wise code generation, which ensures greater accuracy in API usage at each step and reduces the likelihood of API hallucination—a common issue in end-to-end generation baseline approaches.
> Specifically, the case study results show that DocPrompting failed due to its inefficiency in recalling relevant API docs and overlook some important function implementation. CAPIR+Self-Repair struggles to correct the code because its initial solution hallucinates the API usage and deviates significantly from the canonical implementation. And ExploraCoder successfully obtain a correct API usage knowledge from CoAE trace, helping it to generate correct final solution;
>
> *Update: For your convenient reference, we now also put the same case studies content in our [manuscripts' anonymous link](https://anonymous.4open.science/r/ExploraCoder_paper/example.md)

---

> ### Author Response · Authors · 2024-11-24
> **Q2: Additional Explanations on the Dataset Construction Details**
>
> “*how the example programming tasks was created?*”
>
> We observed and analyzed the programming problems in Torchdata-Github and manually integrated the functional requirements of several tasks while ensuring logical consistency. By combining relatively simple, real-world programming tasks to construct more complex example tasks, we believe that these examples are both meaningful and representative.
>
> “*how the expert review was done?*”
>
> We ask the experts to examine on 4 aspect of the crafted programming problems (1) The **executability** of the canonical solution. (2) The **intuitiveness** of the API usage. (3) The **rigor** of the test cases. (3) The **meaningfulness** of the task requirements.
>
> If any issues were identified in these aspects, the experts discussed them with the task creators and revised the tasks accordingly.
>
> We have already updated these additional explanations into the manuscript **Appendix A.5**.

---

> ### Author Response · Authors · 2024-11-24
> **Q3 & Q4: Typos and Content Organization**
>
> We sincerely appreciate your advice, we have corrected all the typos in the paper and will carefully examine the logic flow and provide a rearranged version in the camera-ready.

---

> ### Author Response · Authors · 2024-11-29
> **Looking forward to further feedback**
>
> Dear Reviewer XmiU,
>
> Thank you again for the great efforts and the valuable comments. We have carefully addressed the main concerns in detail and updated the manuscript accordingly. We hope you might find the response satisfactory.
> We believe that the **additional clarification of contribution**, **experiments on extended benchmark**, **analysis on computational overhead**, etc. significantly improve the quality of our submission. If our response has addressed some of your concerns, we sincerely hope you will consider raising the score.
>
> We understand that this is a busy period, and reviewing papers requires a lot of time and effort. As the end of the discussion approaches, we are eager to hear more about your comments on our manuscript and our response. We are happy to clarify any further concerns (if any).
>
> Best Regards,
>
> Authors

---

> > ### Comment · Reviewer_XmiU · 2024-12-01
> >
> > Thank you for your detailed responses to the concerns raised in my review. I genuinely appreciate the substantial effort you have put into the response.
> >
> > While your responses were thorough, I still have fundamental concerns regarding this paper:
> >
> > 1. On novelty: While I acknowledge your effort in distinguishing ExploraCoder from existing code agents, I remain unconvinced that the distinction is sufficient. Generating code for unseen APIs, though different from bug fixing, still falls within the expected capabilities of a general code agent. The methodological differences you highlighted do not represent a fundamental advancement that would justify publication at ICLR.
> >
> > 2. On benchmark: Your justification for Torchdata actually raises several concerns:
> >
> >     a. You argue for "limited pretraining exposure," but Torchdata was released over two years ago. While previous work might have justified using it as an "unseen" API benchmark at the time of their publications, this argument is significantly weaker now. The fast-evolving nature of the code means we need truly current benchmarks.
> >
> >     b. The "Rich and Diverse APIs" claim lacks quantitative support. Without concrete metrics comparing API diversity against other libraries, this remains an unsubstantiated assertion.
> >
> >     c. The "Well-Maintained Documentation" aspect actually weakens your "Representativeness of Torchdata as a Benchmark." Real-world scenarios often involve new APIs with often evolving documentation. By restricting evaluation to a well-documented library, you're testing under idealistic conditions that don't reflect typical challenges.
> >
> > While I appreciate authors' detailed responses a lot and I believe this work contains valuable ideas, I don't think the paper is at a ready state to be accepted at ICLR. I would encourage authors considering addressing the issues in a future revision.

---

### Official Review · Reviewer_vaKZ · 2024-11-03

**Soundness:** 3
**Presentation:** 3
**Contribution:** 2
**Rating:** 6
**Confidence:** 3

**Summary:**

This paper introduces ExploraCoder, a framework for code generation that enables large language models to handle tasks involving multiple, previously unseen APIs. Traditional code generation models, even with RAG or API-pretrained knowledge, often struggle with complex tasks that require multiple API interactions due to limitations in retrieving and integrating diverse APIs effectively. Experiments demonstrate that ExploraCoder significantly outperforms traditional methods.

**Strengths:**

1.The results demonstrate significant improvements, and the experiments are thorough, covering comparisons with RAG frameworks and API-pretrained models across challenging benchmarks.

2.ExploraCoder operates without additional training, making it resource-efficient and easier to deploy, which is advantageous for real-world applications.

3.Clear and Well-Written Presentation: The paper is well-structured and clearly written, making the concepts and experimental setup easy to understand.

**Weaknesses:**

1.The iterative API exploration and self-debugging processes increase computational costs, especially for tasks with complex requirements and multiple APIs. While the framework is training-free, the runtime cost for each exploration cycle could be considerable, impacting scalability for larger applications (eg. repo-level?).

2.I understand that creating datasets is expensive, but I am still concerned about the effectiveness of this method on other libraries. Although the paper demonstrates strong results on the Torchdata-Github and Torchdata-Manual benchmarks, without testing on a more diverse set of libraries, the robustness of ExploraCoder in real-world code generation tasks remains uncertain.

3.ExploraCoder relies on timely execution feedback to guide subsequent steps in the API exploration chain. If execution feedback is delayed or unavailable, as might be the case in certain API-limited or high-latency environments, the framework’s performance could be negatively affected.

4.The framework is designed to handle multiple unseen APIs, but it does not address how it would adapt to changes in APIs over time. As APIs evolve (e.g., deprecations, new parameters, or altered behavior), the effectiveness of previously explored solutions may degrade. Without dynamic updating, ExploraCoder could struggle with outdated API knowledge.

5.The framework’s iterative exploration and debugging processes may produce opaque reasoning steps, making it difficult to understand or verify why certain API calls were chosen. This could hinder debugging or modifying generated solutions manually, which may be problematic for developers aiming to understand the rationale behind each API invocation in the final solution.

6.As mentioned in the paper, if an LLM has already learned new APIs, it would be highly competitive. How can you justify that this method remains competitive when compared to knowledge-update approaches?

7.(minor point) Naive RAG seems like a relatively weak baseline for complex multi-API tasks. Is there no better option to demonstrate that the method in this paper is powerful?

8.(minor point) The method appears straightforward and intuitive, but such process-oriented approaches seem to lack transparency, leading to potential uncontrollable risks in intermediate steps. How do you ensure that the intermediate outputs are reasonable?

**Questions:**

1.Given that iterative API exploration and self-debugging increase computational costs, how does this framework handle large-scale tasks, such as those at the repository level? Have you conducted any efficiency tests or optimizations for such large-scale applications?

2.While the method performs well on the Torchdata benchmarks, how confident are you in its effectiveness for other libraries or domains?

3.If an LLM has already been updated to include new API knowledge, it would naturally perform well. Can you elaborate on how ExploraCoder remains competitive against knowledge-updated models and whether it provides any unique advantages?

4.Are there more suitable baselines that could demonstrate the strengths of ExploraCoder more effectively?

---

> ### Author Response · Authors · 2024-11-24
> **Response Summarization**
>
> Dear reviewer vaKZ, we deeply appreciate your acknowledgment and support of our work. Below, we respond to all the raised weaknesses (W) and questions (Q) with the following organizations:
>
> 1. W1 & Q1: Analysis formula of Computational Costs and Scalability of Application
> 2. W2 & Q2: Scalability of the Dataset and Generalizability of ExploraCoder
> 3. W3: Clarification on Misunderstandings of Performance Decay in Latency Environments
> 4. W4: Clarification on Misunderstandings of Inability in Handling Outdated APIs
> 5. W5: Interpretability of ExploraCoder Reasoning
> 6. W6 & Q3: ExploraCoder with LLM Updated with API knowledge
> 7. W7 & Q4: Clarification on misunderstandings of our baselines
> 8. W8: Ensuring Reasonable Intermediate Outputs in ExploraCoder

---

> ### Author Response · Authors · 2024-11-24
> **W1 & Q1: Analysis Formula of Computational Costs and Scalability of Application**
>
> ## Analysis of Computational Costs
>
> We would like to address the concern you raised in “*the increase computational costs for multiple APIs….impacting the scalability for larger application such as repo-level*”, We first present a detailed breakdown of the computational overhead (model call and token consumption) and corresponding performance for our method and the baselines in tables below(calculation details are provided at the end of our reply). We can observe that：
>
> 1. **Computational costs is proportional to the decomposed subtask (n)**. The decomposition process is also influenced by the number of API invocations required.
> 2. **ExploraCoder is cost-efficient.** When Compared to CAPIR+Self-Repair, ExploraCoder achieves higher performance with fewer model calls and lower token consumption. Additionally, ExploraCoder* achieves a 54% performance improvement over CAPIR+Self-Repair with a 39% increase in token consumption and a 44% increase in model calls, demonstrating that the performance gains significantly outweigh the proportional increase in resource consumption.
> 3. **The overall Computational costs are acceptable:** Our experiments encompass scenarios with varying levels of complexity, including cases requiring 3–14 API invocations. These scenarios represent highly complex real-world development tasks and exceed the complexity of many existing API-related datasets, including some repository-level code generation datasets such as RepoEval.
>
>     To provide intuitive understanding, in a complex task requiring 10 API invocations, ExploraCoder decomposed it into 8 subtasks, involving 19 model calls and consumes 56545 tokens, and ExploraCoder* involves 32 model calls and 95597 token consumption. This level of computational cost is acceptable and comparable to what is commonly observed in other complex code generation tasks according to statistics in [1].
>
>
> |  | model call(pre-processing) |  model call(code generation) | model call(total) | token per sample | pass@10 |
> | --- | --- | --- | --- | --- | --- |
> | naiveRAG | 0 | 1 | 1 | 9932 | 7.81 |
> | EpiGen | n+2 | 1 | 3+n | 18627 | 10.49 |
> | CAPIR | n+2 | 1 | 3+n | 18524 | 12.19 |
> | ExploraCoder | n+2 | n+1 | 3+2n | 56545 | 16.64 |
> | CAPIR + self-repair | n+2 | 1.5n+2 | 4+2.5n | 68970 | 14.62 |
> | ExploraCoder* | n+2 | 2.6n+1 | 3+3.6n | 95597 | 22.19 |
>
> ## Applicability to Larger Repo-Level Scenarios
>
> In our experiments, ExploraCoder primarily focuses on function-level code generation. However, it is also capable of scaling to larger repo-level applications.
>
> The increased complexity of repo-level code generation can be attributed to two main factors:
>
> 1. **More complex functional requirements:** Repo-level tasks often involve generating code across multiple files and functions. These tasks can be decomposed into multiple function-level code generation subtasks, each handled independently by ExploraCoder. In this way, the computational overhead for each function remains manageable and well within the context length limits of current LLMs.
> 2. **Larger API pools:** While repo-level tasks may involve a larger search space of APIs, this increase does not impact the computational complexity of ExploraCoder, since the retrieving process does not involve model call.
>
> Therefore, theoretically, our ExploraCoder could be scaled to larger repo-level applications.
>
> ---
>
>
> ***calculation details of the table**
>
> For model calls, we provide clear analytical expressions based on n, the number of decomposed subtasks. For token consumption, we randomly sampled 10 tasks with n=8 (the mean value of n in our dataset), generated 20 candidate final solutions, and calculated the average token consumption per task.
>
> For the two approaches involving self-debug mechanisms, Self-Repair and ExploraCoder*, debugging rounds is not deterministic. Theirfore we use the formulated expectation based on the empirically observed debug rate in our experiments. Let’s set the probability of the two methods conducting debug as $p_1$ (ExploraCoder*) and $p_2$ (Self-repair). Their expectation of model call can be formulated as $(1+5p_1)n+1$ and $2p_2n+2$. Notably, $p_2$=0.75 is significantly higher than $p_1$=0.32 across two benchmark. This debug rate difference arises because ExploraCoder* focuses on debugging simple subtasks (which are generally less error-prone), while Self-Repair always attempts to debug a complete solutions (which often fail to repair successfully, triggering additional debugging iterations up to the budget limit).
>
> [1] Agentless: Demystifying LLM-based Software Engineering Agents. Xia et al. https://arxiv.org/abs/2407.01489

---

> ### Author Response · Authors · 2024-11-24
> **W4: Scalability of the Dataset and Generalizability of ExploraCoder**
>
> ## 1. The Justification for Torchdata-Library-Based Evaluation
>
> We‘d like to address your concern about the Torchdata-Library-Based Evaluation and the generalizability of ExploraCoder to other libraries or domain.
>
> ### 1.1 Torchdata-based evaluation aligns with prior work
>
> Firstly, we would like to clarify the choice of our Torchdata-based benchmark. Our selection aligns with prior work on unseen-library tasks [1][2][3][4] and does not imply a specific preference for the Torchdata library. Torchdata was selected due to its unique combination of qualities, which are currently **difficult to identify in other open-source libraries**:
>
> 1. **Limited Pretraining Exposure**, making it suitable for conducting evaluation on unseen API invocation tasks.
> 2. **Rich and Diverse APIs**, making it suitable for constructing a multi-API benchmark with diverse and meaningful evaluations.
> 3. **Well-Maintained Documentation,** making it particularly suitable for retrieval-based code generation methods.
>
> ### 1.2 Representativeness of Torchdata as a Benchmark
>
> Secondly, **Torchdata holds a certain level of representativeness among external libraries.** Torchdata provides practical data processing APIs for Pytorch, which is objectively and functionally similar to many other commonly used libraries such as Pandas, Numpy, Huggingface Datasets etc.(Unfortunately, their API knowledge has already been exposed to advanced LLMs), and could resemble many privately-maintained or up-coming libraries. Therefore, we believe that evaluating ExploraCoder on Torchdata serves as a meaningful reference for its applicability to other libraries.
>
> ### 1.3 Enhancing Diversity and Generalizability in the Torchdata Benchmark
>
> Moreover, to enhance the **diversity and generalizability** of our Torchdata-based benchmark, we also sampled a variety of API invocations and combinations from the Torchdata API pool as mentioned in data construction section in our original manuscript Appendix A.5.
>
> Therefore, we consider our benchmark to be a valuable and meaningful resource for evaluating LLM’s ability on unseen-library code generation.
>
> ### 1.4 ExploraCoder is Torchdata-Agnostic
>
> Additionally, our proposed **ExploraCoder is general plug-and-play framework** and does not rely on any Torchdata-specific steps. As such, it **can be applied to other library’s programming problems.**
>
> ## 2. The Benchmark is now Extended
>
> To further address the your concerns regarding the diversity and generalizability of our benchmark, we have expanded the Torchdata-Manual dataset. Using the same methodology as before, we manually curated an additional 50 complex programming problems that **involves distinct API compositions from the original dataset**. As a result, our Torchdata-based benchmark now contains 150 diversed programming problems.
>
> We have conducted experiments on representative baselines using GPT-3.5-turbo (the primary base model used in our original paper), and the results are presented below:
>
> |  | pass@1 | success@1 | pass@5 | success@5 | pass@10 | success@10 | pass@20 | success@20 |
> | --- | --- | --- | --- | --- | --- | --- | --- | --- |
> | direct | 0 | 0 | 0 | 0 | 0 | 0 | 0 | 0 |
> | naive RAG | 0.28 | 0.75 | 1.31 | 3.64 | 2.38 | 7.05 | 3.72 | 13.16 |
> | CAPIR | 3.26 | 5.78 | 8.33 | 10.81 | 10.64 | 17.41 | 13.33 | 26.70 |
> | ExploraCoder | 9.43 | 19.89 | 14.38 | 27.17 | 16.07 | 28.35 | 17.55 | 29.32 |
> | CAPIR + Self-Repair | 9.61 | 16.52 | 9.80 | 19.87 | 9.80 | 21.26 | 9.80 | 23.32 |
> | ExploraCoder* | 14.24 | 27.22 | 22.60 | 40.21 | 24.99 | 43.71 | 27.22 | 46.51 |
>
> The trends observed in the results are consistent with the original paper, with ExploraCoder achieving SOTA performance. This consistency allows us to draw similar analytical conclusions, demonstrating the robustness of our Evaluation. This result is updated in **Appendix A.9**. The remaining experiments in the original manuscript will be conducted on the extended dataset, and the results will be updated into the manuscripts.
>
> ---
>
> [1] When Language Model Meets Private Library. Zan et al. https://arxiv.org/abs/2210.17236
>
> [2] Private-Library-Oriented Code Generation with Large Language Models. Zan et al. https://arxiv.org/abs/2307.15370
>
> [3] Compositional API Recommendation for Library-Oriented Code Generation. Ma et al. https://arxiv.org/abs/2402.19431
>
> [4] ToolCoder: Teach Code Generation Models to use API search tools. Zhang et al. https://arxiv.org/abs/2305.04032

---

> ### Author Response · Authors · 2024-11-24
> **W3: Clarification on Misunderstandings of performance decay in latency environments**
>
> We would like to clarify that **our method is not negatively affected under the mentioned conditions.**
>
> (1) high-latency environment: ExploraCoder adopts a synchronous waiting strategy for each intermediate API invocation. This means that if network latency occurs, the framework waits for the normal response before proceeding to the next step. As a result, latency does not lead to errors in the execution or final outcomes of our method.
>
> (2) API-limited environment: If the reviewer’s reference to “API-limited environments” implies scenarios with restricted model calls, we would like to note that resolving the availability of LLMs is outside the scope of this work. The invocation of LLMs is transparent to our framework. Therefore, in such environments, users can employ any available LLM to suit their needs without impacting ExploraCoder’s functionality.

---

> ### Author Response · Authors · 2024-11-24
> **W4: Clarification on Misunderstandings of Inability in Handling Outdated APIs**
>
> ExploraCoder generate instant solution based on a continuously updated API pool. Given a requirement, ExploraCoder dynamically retrieve the latest API doc and explore on the API usage, and finally generates the code solution. **Therefore there’s no existence of outdated solution.** It is important to note that the maintenance and updating of the API pool is an easily automatable process and falls outside the scope of this work.

---

> ### Author Response · Authors · 2024-11-24
> **W5: Interpretability of ExploraCoder Reasoning**
>
> We would like to clarify that ExploraCoder provides traceable and transparent reasoning through the following features:
>
> 1.	**Logged exploration process:** ExploraCoder logs every intermediate step, including the generated code snippets and their corresponding execution results.
>
> 2.	**Semantic clarity of reasoning subtasks:** Each reasoning subtask in the exploration process is explicitly described with clear semantic meaning, aiding understanding.
>
> 3.	**Trackable API recommendations:** The APIs recommended for each intermediate subtask are selected based on their similarity to the subtask description, and this process is fully traceable.
>
> 4.	**Error tracking for debugging:** Intermediate steps that result in execution errors are also logged, allowing developers to trace and debug effectively.
>
> By combining these features, ExploraCoder provides the traceability and interpretability from the final solution back to each intermediate step, empowering developers to understand the rationale behind ExploraCoder.
>
> We also provided a case study of ExploraCoder* in Listing 11 of the original manuscript, where we demonstrate its exploration trace in detail.

---

> ### Author Response · Authors · 2024-11-24
> **W6 & Q3: ExploraCoder on LLM with Updated API Knowledge**
>
> We appreciate the reviewer’s question regarding ExploraCoder’s competitiveness against LLMs updated with API knowledge. In fact, our experiments have already discussed this scenario. Specifically, we evaluated **GPT-4-1106-preview**, an updated version of GPT-4 with knowledge of the Torchdata library, as a knowledge-updated counterpart to **GPT-4-0613**, which lacks prior Torchdata knowledge. The results, as summarized in the table below, reveal the following insights:
>
> 1.	**Knowledge-updated LLMs naturally perform better:** As expected, LLMs with API knowledge (e.g., GPT-4-1106-preview) generally outperform those without prior knowledge in all settings. This demonstrates the advantage of incorporating updated API knowledge into the model.
>
> 2.	**ExploraCoder w/o prior knowledge is still highly competitive even against knowledge-updated LLMs:** In our experiments, a no-prior-knowledge model combined with ExploraCoder outperformed a knowledge-updated model enhanced with RAG. This is due to the following reasons:
>
> - **Complexity of multi-API invocation tasks:** These tasks require not only API knowledge but also strong reasoning abilities to coordinate multiple API invocations effectively.
> - **Lack for complete API usage knowledge:** Even in knowledge-updated models, API-related information is often sparse, and publicly available API documentation is typically incomplete, especially for the usage when interacting with other APIs.
>
> 3.	**ExploraCoder is also beneficial for knowledge-updated LLMs:** Beyond its effectiveness with no-prior-knowledge models, ExploraCoder can significantly enhance the performance of knowledge-updated LLMs. This is achieved through its **CoAE (Chained API Exploration)** process, which dynamically gathers information about API interactions and generates accurate API invocation chains step by step. By doing so, ExploraCoder complements the inherent limitations of LLMs, even those equipped with updated API knowledge, by providing a robust mechanism for reasoning and exploration.
>
> | base model | method | pass@20 | success@20 |
> | --- | --- | --- | --- |
> | GPT-4-0613 (w/o prior knowledge)  | direct | 0 | 0 |
> | | + RAG | 0 | 7.71 |
> |  | + ExploraCoder | 27.71 | 45.70 |
> | GPT-4-1106-preview (knowledge-updated) | direct | 3.80 | 21.05 |
> |  | + RAG | 13.90 | 27.90 |
> |  | + ExploraCoder | 33.71 | 63.10 |

---

> ### Author Response · Authors · 2024-11-24
> **W7 & Q4: Clarification on misunderstandings of our baselines**
>
> We'd like to clarify your concern in “Naive RAG seems like a relatively weak baseline… Is there no better option…?”.
>
> This maybe a potential misunderstanding, actually we provide a progressive comparison from naive RAG (Table 1-2) to more advanced RAG approaches(Table 5-6).  We selected two SOTA library-oriented code generation approaches EpiGen and CAPIR, and crafted a stronger baseline by ioncorporating CAPIR with SOTA debug mechnism, Self-Repair. The experimental results clearly demonstrate the superiority of ExploraCoder over all strong baselines in terms of both pass rate and success rate.
>
> For better comprehension, we **updated** and emphasized the progressive comparison design in the caption of **Table 1-2** to guide readers in understanding the structure of our experiments and the rationale behind our baseline selection.

---

> ### Author Response · Authors · 2024-11-24
> **W8: Ensuring Reasonable Intermediate Outputs in ExploraCoder**
>
> Firstly, the intermediate reasoning process in ExploraCoder is fully transparent, allowing users to observe and trace all steps in real-time. Any unreasonable planning or erroneous intermediate outputs are logged and made visible, ensuring developers have complete visibility into the system’s reasoning.
>
> Secondly, while ExploraCoder is currently designed to operate in an end-to-end manner to minimize human effort, it can be easily adapted into a “human-in-the-loop” workflow for greater controllability if desired. This flexibility allows developers to intervene and adjust the intermediate outputs as needed, effectively balancing automation and control.

---

> ### Comment · Reviewer_vaKZ · 2024-11-25
>
> Thank you for your detailed rebuttal and for addressing some of my concerns. I appreciate the effort and thoughtfulness you put into resolving these issues, which has led me to raise my score to 6.

---

> ### Author Response · Authors · 2024-11-25
>
> We deeply appreciate your acknowledgment and the decision to increase the score. Once again, thank you very much for all your constructive suggestions!

---

### Official Review · Reviewer_oLK8 · 2024-11-04

**Soundness:** 3
**Presentation:** 2
**Contribution:** 3
**Rating:** 6
**Confidence:** 2

**Summary:**

The paper introduces ExploraCoder, a training-free framework designed for large language models (LLMs) to utilize unseen APIs to perform code generation (library-oriented code generation), enabling LLMs’ capabilities to adapt to new or private libraries without retraining.
ExploraCoder decomposes a complex programming task into simpler subtasks, employing a chain-of-API-exploration (CoAE) method that iteratively generates, executes, and refines code snippets for specific API calls for each subtask.
By doing so, ExploraCoder builds an experience trace, ultimately guiding the generation of correct final code.
The paper demonstrates improved performance in handling unseen APIs on the Torchdata-based benchmarks compared to other training-free methods, achieving competitive Pass@k and Success@k rates.

**Strengths:**

1. The problem of adapting LLMs to work with new APIs is a timely and important as it reflects the real-world need for adaptability in code generation, especially in environments with dynamic or proprietary libraries. Chain of API Exploration is innovative in the context of LLMs.
2. The framework for handling new APIs -- divide-and-conquer framework combined with iterative API exploration which mimics a human programmer’s exploratory approach is solid and well-supported. Interestingly, the self-debugging mechanism is well-integrated within CoAE.
3. The experimental design appears thorough, incorporating both existing and newly developed benchmarks to assess performance across tasks of varying complexity. The benchmarks are carefully constructed to highlight the challenges of multi-API invocation.
4. Results are clearly presented, with ablation studies and comparisons to relevant baselines (e.g., naive RAG, Self-Repair) supporting claims of improved performance

**Weaknesses:**

1. The use of similarity-based retrieval strategy for identifying appropriate APIs for subtasks may not be effective in case APIs are similar in functionality functionally  but syntactically different (e.g., APIs that share functionalities across domains or are named differently). This inherits the retrieval bias, which may hinder the subtask performance.

2. ExploraCoder may be sensitive to effective task segmentation, or selecting the optimal granularity for subtasks, where over- or under-decomposition may lead to errors or inefficiencies. The paper lacks a proper discussion and evaluation of how accurately they achieve optimal granularity.

* Furthermore, the paper does not specify how the model handles cascading errors in multi-API tasks, especially for the
iterative CoAE mechanism and self-debugging.

3. While the paper sometimes implies that ExploraCoder can generalize across various API types, the benchmarks are limited to the Torchdata library, which may not fully represent and generalize to APIs in other domains.

**Typos and Minor Errors**:
   - Page 1, Abstract: "niave" should be "naive."
   - Page 2, Section 1: "self-debug" is not consistently hyphenated throughout the paper.
   - Terms like "experience exploitation" and "trace" are not defined until later sections, which could confuse readers initially.

**Questions:**

1. How does the framework handle APIs with similar functionality but different syntax? How robust is the similarity-based retrieval in this setting?
2. Have you evaluated different segmentation strategies, and how did they affect model performance?
3. How often the framework must retry or debug when errors accumulate across tasks?

---

> ### Author Response · Authors · 2024-11-24
> **Response Summarization**
>
> Dear reviewer oLK8, we deeply appreciate your acknowledgment and support of our work. Below, we respond to all the raised weaknesses (W) and questions (Q) with the following organizations:
>
> 1. W1 & Q1: The Effectiveness of Similarity-based Retrieval
> 2. W2: The Effectiveness of our Task Segmentation
> 3. W3: Clarification for the Cascading Errors
> 4. W4: Scalability of the Dataset and Generalizability of ExploraCoder
>     1. The Justification for Torchdata-Library-Based Evaluation
>     2. The Benchmark is now Extended
> 5. Q2: Model Performance Under Different Segmentation Strategies
> 6. Q3: How often does ExploraCoder debug

---

> ### Author Response · Authors · 2024-11-24
> **W1 & Q1: The effectiveness of similarity-based retrieval**
>
> We appreciate the reviewer’s concern about the robustness of our similarity-based retrieval strategy in handling APIs with similar semantics but different syntax. Below, we address this concern:
>
> 1. **Use of LLM-based embeddings for robust semantic understanding:**
>
>     Our method utilizes the **text-embedding-ada-002** model to compute vector representations of API documentation and subtask descriptions. This embedding model is designed to encode semantically similar content into similar vectors, regardless of syntactic differences—even across different languages, as demonstrated in tasks such as machine translation [1]. As a result, the impact of syntactic variations on our retrieval effectiveness is minimal.
>
> 2. **Empirical evidence from our dataset:**
>
>     Our dataset includes scenarios where APIs have significant syntactic differences but share similar functionalities. For example:
>
>     - Given the subtask description: “Process the data to extract specific columns using extract_columns_map_fn”
>
>     - Our retriever successfully retrieved the canonical API: “torchdata.datapipes.map.Mapper: Apply the input function over each item from the source DataPipe.”
>
>     The similarity score was 77%, showing that the embedding model effectively captured the functional/semantic similarity between **“Process xxx to”** and **“Map”**, despite their syntactic differences.
>
>
> These results demonstrate that our retrieval strategy, based on LLM-generated embeddings, is robust to syntactic variations and capable of identifying functionally appropriate APIs. This capability ensures that our framework remains effective even when APIs exhibit significant syntax differences.
>
> [1] An Empirical Study of In-context Learning in LLMs for Machine Translation. Chitale et al. https://arxiv.org/abs/2401.12097

---

> ### Author Response · Authors · 2024-11-24
> **W2: The effectiveness of our task segmentation**
>
> We sincerely appreciate that you pointing out the lack for “a proper discussion and evaluation of how accurately they achieve task segmentation optimal granularity.”
>
> Although It is hard to directly quantify the quality of decomposition granularity, we can **evaluate it indirectly by calculating the number of APIs included in each subtask**, as discussed in Section 3.2 line 175 in our original manuscript, our design **aims to ensure each decomposed subtask involves 1-2 API explorations**, so that it's easy enough to be solved.
>
> As shown in Table below, the average number of decomposed subtasks by GPT-3.5 is closely aligned with the average number of API invloved across two datasets. This indicates that the decomposition strategy effectively achieves the desired granularity. The ExploraCoder’s overall performance also indicates the effectiveness of our task segmenration. This result is updated in **Appendix A.8**.
>
> |  | #API invocation | #decomposed subtask | API per subtask |
> | --- | --- | --- | --- |
> | Torchdata-Github | 4.26 | 4.06 | 1.04 |
> | Torchdata-Manual | 9.94 | 8.22 | 1.21 |

---

> ### Author Response · Authors · 2024-11-24
> **W3: Clarification for the cascading errors**
>
> In Section 5.3 of our original manuscript, we have discussed the cascading errors condition:
> > “intuitively, the failures in the exploration chain, such as wrong API usage could propagate through API interactions and affect the generation of final solutions.“
>
> To address this issue, we introduced a variant, **ExploraCoder***, which prompts the LLM to self-debug when an intermediate subtask fails. By fixing errors in intermediate steps in real-time, this mechanism prevents the propagation of erroneous or “poisonous” API usage knowledge along the exploration path. Our experimental results in Table 4 of original manuscript demonstrate the effectiveness of this approach, showcasing significant improvements in handling cascading errors.

---

> ### Author Response · Authors · 2024-11-24
> **W4: Scalability of the Dataset and Generalizability of ExploraCoder**
>
> ## 1. The Justification for Torchdata-Library-Based Evaluation
>
> We‘d like to address your concern about the evaluation in comment “the benchmarks are limited to the Torchdata library, which may not fully represent and generalize to APIs in other domains.”
>
> ### 1.1 Torchdata-based evaluation aligns with prior work
>
> Firstly, we would like to clarify the choice of our Torchdata-based benchmark. Our selection aligns with prior work on unseen-library tasks [1][2][3][4] and does not imply a specific preference for the Torchdata library. Torchdata was selected due to its unique combination of qualities, which are currently **difficult to identify in other open-source libraries**:
>
> 1. **Limited Pretraining Exposure**, making it suitable for conducting evaluation on unseen API invocation tasks.
> 2. **Rich and Diverse APIs**, making it suitable for constructing a multi-API benchmark with diverse and meaningful evaluations.
> 3. **Well-Maintained Documentation,** making it particularly suitable for retrieval-based code generation methods.
>
> ### 1.2 Representativeness of Torchdata as a Benchmark
>
> Secondly, **Torchdata holds a certain level of representativeness among external libraries.** Torchdata provides practical data processing APIs for Pytorch, which is objectively and functionally similar to many other commonly used libraries such as Pandas, Numpy, Huggingface Datasets etc.(Unfortunately, their API knowledge has already been exposed to advanced LLMs), and could resemble many privately-maintained or up-coming libraries. Therefore, we believe that evaluating ExploraCoder on Torchdata serves as a meaningful reference for its applicability to other libraries.
>
> ### 1.3 Enhancing Diversity and Generalizability in the Torchdata Benchmark
>
> Moreover, to enhance the **diversity and generalizability** of our Torchdata-based benchmark, we also sampled a variety of API invocations and combinations from the Torchdata API pool as mentioned in data construction section in our original manuscript Appendix A.5.
>
> Therefore, we consider our benchmark to be a valuable and meaningful resource for evaluating LLM’s ability on unseen-library code generation.
>
> ### 1.4 ExploraCoder is Torchdata-Agnostic
>
> Additionally, our proposed **ExploraCoder is general plug-and-play framework** and does not rely on any Torchdata-specific steps. As such, it **can be applied to other library’s programming problems.**
>
> ## 2. The benchmark is now extended
>
> To further address the your concerns regarding the diversity and generalizability of our benchmark, we have expanded the Torchdata-Manual dataset. Using the same methodology as before, we manually curated an additional 50 complex programming problems that **involves distinct API compositions from the original dataset**. As a result, our Torchdata-based benchmark now contains 150 diversed programming problems.
>
> We have conducted experiments on representative baselines using GPT-3.5-turbo (the primary base model used in our original paper), and the results are presented below:
>
> |  | pass@1 | success@1 | pass@5 | success@5 | pass@10 | success@10 | pass@20 | success@20 |
> | --- | --- | --- | --- | --- | --- | --- | --- | --- |
> | direct | 0 | 0 | 0 | 0 | 0 | 0 | 0 | 0 |
> | naive RAG | 0.28 | 0.75 | 1.31 | 3.64 | 2.38 | 7.05 | 3.72 | 13.16 |
> | CAPIR | 3.26 | 5.78 | 8.33 | 10.81 | 10.64 | 17.41 | 13.33 | 26.70 |
> | ExploraCoder | 9.43 | 19.89 | 14.38 | 27.17 | 16.07 | 28.35 | 17.55 | 29.32 |
> | CAPIR+self-repair | 9.61 | 16.52 | 9.80 | 19.87 | 9.80 | 21.26 | 9.80 | 23.32 |
> | ExploraCoder* | 14.24 | 27.22 | 22.60 | 40.21 | 24.99 | 43.71 | 27.22 | 46.51 |
>
> The trends observed in the results are consistent with the original paper, with ExploraCoder achieving SOTA performance. This **consistency** allows us to draw similar analytical conclusions, **demonstrating the robustness of our Evaluation**. This result is updated in **Appendix A.9**. The remaining experiments in the original manuscript will be conducted on the extended dataset, and the results will be updated into the manuscripts.
>
> [1] When Language Model Meets Private Library. Zan et al. https://arxiv.org/abs/2210.17236
>
> [2] Private-Library-Oriented Code Generation with Large Language Models. Zan et al. https://arxiv.org/abs/2307.15370
>
> [3] Compositional API Recommendation for Library-Oriented Code Generation. Ma et al. https://arxiv.org/abs/2402.19431
>
> [4] ToolCoder: Teach Code Generation Models to use API search tools. Zhang et al. https://arxiv.org/abs/2305.04032

---

> ### Author Response · Authors · 2024-11-24
> **Q2: Model performance under different segmentation strategies**
>
> “*Have you evaluated different segmentation strategies, and how did they affect model performance?*”
>
> Yes, in our original manuscript’s ablation study, we compare our planner with a variant segmentation strategy where we drop the library doc information in planning phase and solely let LLM to segment the task. The result below shows the effectiveness of our planner and the challenge of planning on unseen library API invocations.
>
> | planner | pass@1 | success@1 |
> | --- | --- | --- |
> | w/ doc (ours) | 8.76 | 15.48 |
> | w/o doc | 3.90 | 15.22 |

---

> > ### Comment · Reviewer_oLK8 · 2024-11-26
> > **Re: Response from Authors**
> >
> > Thank you for your detailed response.
> >
> > * For Q1, to clarify, my question is to understand how (intuitively) the framework handles "APIs with similar functionality but different syntax".
> > Furthermore, if possible, can you please provide examples or case studies to better illustrate this scenario?
> >
> > * For W2, can you elaborate more on how the framework handles "cascading errors in multi-API tasks, especially for the iterative CoAE mechanism and self-debugging"?
> >
> > Best

---

> > > ### Author Response · Authors · 2024-11-29
> > > **(Part 1/2 of) Further Explanation on Q1**
> > >
> > > *ps: We are not entirely certain about the specific scenario you refer to as “APIs with similar functionality but different syntax.” If our response below does not fully resolve your concern, we would greatly appreciate it if you’d give us further clarification and guidance.*
> > >
> > > Based on our careful understanding, we have rephrased your question as: **“How we identify the correct API for subtask among so many different APIs that potentially shares similar functionality”**
> > >
> > > Intuitively, our API recommendation and CoAE modules can be understood as a process that progressively narrows down the promising API search space and identifies the correct API for the final solution generation. This involves three key steps:
> > >
> > > ## 1. Similarity-Based Candidate API Retrieval
> > >
> > > **For each subtask, we retrieve a set of $k > 1$ candidate APIs** to maximize the likelihood that **most functional relevant APIs are recalled** in this coarse-grained similarity-based retrieval. We use dense retrieval to capture the semantical relevancy between subtask and APIs (as detailed in our [previous response](https://openreview.net/forum?id=m5rOrTiuKG&noteId=aLSpvdsPMN)).
> > >
> > > ## 2. LLM-Based API Selection
> > >
> > > In the CoAE process, the LLM further refines the candidate set by attempting to invoke the promising APIs for current subtask implementation. Specifically, all retrieved candidate APIs are fed to the LLM’s context window. By setting the temperature, we can **sample $m$ diverse API invocation snippets from LLM**, each closely adhering to the programming problem’s objectives. Intuitively, **APIs with even higher functional relevancy are selected in this step** based on the LLM’s powerful semantic understanding and reasoning capabilities.
> > >
> > > **Example:**
> > >
> > > In this example, given a recommended set of functionally relevant APIs, the LLM carefully captured the nuanced differences between the APIs by reading their documentation and ultimately selected the more appropriate APIs, `FSSFileLister` and `FSSpecOpener`, for the subtask.
> > >
> > > ```python
> > > # [SUBTASK]: Open and read each text file
> > > # [Recommended APIs]:
> > > torchdata.datapipes.iter.FSSpecFileLister
> > > torchdata.datapipes.iter.IoPathFileOpener
> > > torchdata.datapipes.iter.FSSpecFileOpener
> > > torchdata.datapipes.iter.OnlineReader
> > > torchdata.datapipes.iter.GDriveReader
> > > torchdata.datapipes.iter.HttpReader
> > > torchdata.datapipes.iter.FileOpener
> > > torchdata.datapipes.iter.FileLister
> > > torchdata.datapipes.iter.AISFileLoader
> > > torchdata.datapipes.iter.ZipArchiveLoader
> > >
> > > ## [Candidate API invocation 1]
> > > file_lister = FSSpecFileLister(root=root)
> > > file_opener = FSSpecFileOpener(source_datapipe=file_lister, mode='r')
> > > for file_path, file_stream in file_opener:
> > >     print(f"File Path: {file_path}")
> > >     print("File Content:")
> > >
> > > ## [Candidate API invocation 2]
> > > file_lister = FSSpecFileLister(root=root)
> > > file_opener = FSSpecFileOpener(source_datapipe=file_lister)
> > > for file_path, file_stream in file_opener:
> > >     print(file_path)
> > >     print(file_stream.read())
> > > ...
> > > ```

---

> > > ### Author Response · Authors · 2024-11-29
> > > **(Part 2/2 of) Further Explanation on Q1**
> > >
> > > ## 3. Execution-Based Correct API Identification
> > >
> > > Among the m API invocation snippets generated by the LLM, CoAE prioritize selecting the successfully executable API invocation as the current step in the exploration path. In this process, APIs that are seemingly functionally similar but incompatible with our context will be screened out. The APIs selected along these CoAE paths are parsed and added to the LLM’s context of the final solution generation.
> > >
> > > **Example:**
> > >
> > > In this example, two highly functionally relevant APIs, `Mapper` and `FlatMapper`, were selected by the LLM. While these APIs share similar functionalities, they exhibit some behavioral differences. By passing the output of the `json_parser` from the prior API invocation to both APIs, we observed that `Mapper` successfully executed and solved the subtask, whereas `FlatMapper` was incompatible with the prior API usage and failed to execute. Therefore, in this subtask, **we successfully identified and selected the correct API from two functionally similar options.**
> > >
> > > ```python
> > > # [SUBTASK]: Flatten nested JSON objects using flatten_json_objects
> > >
> > > # [Candidate API invocation 1]:
> > > flattened_json_data = Mapper(datapipe=json_parser, fn=flatten_json_objects)
> > > for file_name, json_data in flattened_json_data:
> > >     print(file_name)
> > >     print(json_data)
> > > ## [Execution Observation]:
> > > ./torchdata-programming-tasks/task_75/file_0.json
> > > [{'field1': 13, 'field2': 5, 'field3': 1}, {'field1': 13, 'field2': 36, 'field3': 32}...]
> > >
> > > ## [Candidate API invocation 2]:
> > > flattened_json_data = FlatMapper(datapipe=json_parser, fn=flatten_json_objects)
> > > for file_name, json_data in flattened_json_data:
> > >     print(file_name)
> > >     print(json_data)
> > > ## [Execution Observation]:
> > > ValueError: too many values to unpack (expected 2)
> > > ```
> > >
> > > ## Moreover, a subtask can be solved by different APIs
> > >
> > > It is common for a specific subtask to be solvable by multiple APIs that share similar functionality but differ in syntax (e.g., API names or usage patterns). Our method accommodates this phenomenon by leveraging execution feedback at each step to understand the behavior of the selected API. This feedback ensures that subsequent subtasks are better aligned with the selected API, even when multiple valid APIs are available.
> > >
> > > **Example:**
> > >
> > > In fact, our dataset includes cases of functionally similar APIs like this. An shown in example below, `FSSpecFileLister` and `FileLister` both can be used to list all files in a given path. However, they are syntactically different and have slight differences in implementation: the former lists the full path of files, while the latter provides only relative paths. In our subtask, both APIs were retrieved and utilized by the LLM to generate two different API invocations for solving the subtask, both of which executed successfully. In such cases, we **randomly selected one of the correct APIs for problem-solving.**
> > >
> > > ```python
> > > # Subtask: List all files under the given root directory
> > >
> > > ## [Candidate API invocation 1]:
> > > files = FSSepcFileLister(root=file_dir)
> > > for file in files:
> > >     print(file)
> > > ## [Execution Observation]:
> > > /home/anonymous/.../torchdata-programming-tasks/task_2/file_0.json
> > > /home/anonymous/.../torchdata-programming-tasks/task_2/file_1.json
> > > /home/anonymous/.../torchdata-programming-tasks/task_2/file_2.json
> > > /home/anonymous/.../torchdata-programming-tasks/task_2/file_3.json
> > > /home/anonymous/.../torchdata-programming-tasks/task_2/file_4.json
> > >
> > > ## [Candidate API invocation 2]:
> > > files = FileLister(root=file_dir)
> > > for file in files:
> > >     print(file)
> > > ## [Execution Observation]:
> > > ./torchdata-programming-tasks/task_2/file_0.json
> > > ./torchdata-programming-tasks/task_2/file_1.json
> > > ./torchdata-programming-tasks/task_2/file_2.json
> > > ./torchdata-programming-tasks/task_2/file_3.json
> > > ./torchdata-programming-tasks/task_2/file_4.json
> > > ```

---

> > > ### Author Response · Authors · 2024-11-29
> > > **(Part 1/2 of) Elaboration on Cascading Error in Multi-API Tasks (W3)**
> > >
> > > In our original manuscript, we discussed the challenges of invoking **multiple unseen APIs** (line 215), noting that “LLMs tend to hallucinate the usage of unfamiliar APIs, and errors can propagate through API interactions.”
> > >
> > > Since we could not find a formal definition for "cascading error" (as indicated in the in your comment) in the context of code generation, to conduct a rigorous dicussion, we now try to define **propagation/cascading errors** in the context of multi-API tasks as: **errors in earlier API usage leading to failures at subsequent API invocations**. Based on this definition, we discuss two types of “cascading errors” empirically observed in our study and demonstrate how our proposed framework mitigates such issues through the CoAE mechanism and self-debugging processes.
> > >
> > > *ps: If there are any misunderstandings or inaccuracies, we would greatly appreciate it if you could kindly point them out. We are willing to revise our discussion and further clarify our expressions.*
> > >
> > >
> > > ## 1. Propagation of API Hallucination in the Decoding Process
> > >
> > > One common type of error occurs in end-to-end generation, where API hallucinations in early decoding step propagate to subsequent API calls due to the autoregressive nature of LLM, resulting in a **pervasive API misuse pattern across the entire code solution**. In our task of multiple unseen API generation, hallucinations are more likely to arise due to LLM’s **insufficient attention** when handling a large number of unfamiliar API information [1].
> > >
> > > We observed one example below of CAPIR misusing all the API calls as class initialization. We find code in such cases not only unusable for developer, but also hard for holistic debugging method such as Self-Repair to fix.
> > >
> > > ```python
> > > ...
> > > dp_files = S3FileLoader(s3_prefixes)()
> > > dp_lines = LineReader(dp_files, strip_newline=True)()
> > > dp_words = Mapper(dp_lines, extract_words_map_fn)()
> > > dp_grouped = Grouper(dp_words, group_key_fn, group_size=10000)()
> > > dp_combined = Mapper(dp_grouped, combine_words_map_fn)()
> > > ...
> > >
> > > TypeError: 'S3FileLoaderIterDataPipe' object is not callable
> > > ```
> > >
> > > **ExploraCoder mitigates the pervasive misuse pattern through CoAE:**
> > >
> > > - **Focused Step-wise Generation:** Each intermediate subtask involves exploring only a small number of unfamiliar APIs, leading to more accurate API invocations generation.
> > > - **Selecting Correct API Usage:** For every subtask, ExploraCoder prioritizes selecting API invocations that execute successfully, and accumulates this knowledge in the context as long term memory. Successful prior subtasks in context not only mitigate cascading errors but also propagate correct API usage patterns to subsequent subtasks.
> > >
> > > ---
> > > [1] A Survey on Hallucination in Large Language Models: Principles, Taxonomy, Challenges, and Open Questions. Huang et al. https://arxiv.org/abs/2311.05232

---

> ### Author Response · Authors · 2024-11-29
> **(Part 2/2 of) Elaboration on Cascading Error in Multi-API Tasks (W3)**
>
> ## 2. Propagation of API Invocation Failures in Sequential Subtasks
>
> In multi-API tasks, subsequent API invocations often depend on the outputs of earlier APIs. A failure in dependent API invocation could directly lead to failures of subsequent API in performing their functionality, regardless of whether the subsequent API usage is correct. This could manifest as a cascading error in multi-step pipelines like ExploraCoder, where failures in earlier unresolved subtasks can disrupt the entire workflow.
>
> **We mitigate such issue from two aspects of design:**
>
> ### **(1) Loosen the Coupling Between Steps:**
> In CoAE, we do not force LLM to incrementally generate based soley on prior subtask’s output, instead, we ask LLM to take prior API invocations and their execution feedback as references/experiences. At each step, LLM can flexibly reuse or reorganize the prior API invocation code to build the necessary test inputs for current subtask. In the final solution generation phase, the loose-coupled subtasks implementations which contain rich API usage knowledge will be gathered again for an integral code generation, where some of the decouple API dependency could be rebuilt.
>
> In the following example, we observe that while a prior dependent batch-processing subtask failed, LLM in the later subtask decided to skip the ”BatcherMapper“ failure, and directly conduct the current subtask by enumerating on the json object. This allowed the behaviour of “Enumerator” API to be successfully observed, ensuring that the subsequent workflow of CoAE continuously and successfully proceeded without cascading errors.
>
> ```python
> # Task: load JSON data from files and process the data in batch using process_batch_fn, then enumerate the batch with indices...
> ...
> ## [Subtask 4]: Process the JSON object data in batch using process_batch_fn
> file_lister = FileLister(root=root)
> json_opener = FileOpener(file_lister)
> json_parser = JsonParser(json_opener)
> batch_mapper = BatchMapper(json_parser, fn=process_batch_fn, batch_size=batch_size)
> ## [Execution Observation]
> TypeError: 'str' object does not support item assignment
>
> ## [Subtask 5]: Enumerate the data with indices
> file_lister = FileLister(root=root)
> file_opener = FileOpener(file_lister)
> json_parser = JsonParser(file_opener)
> enumerator = Enumerator(json_parser)
> for index, (file_path, json_content) in enumerator:
>     print(f"Index: {index}")
>     print(f"File path: {file_path}")
>     print(f"JSON content: {json_content}")
> ## [Execution Observation]
> Index: 0
> File path: ./torchdata-programming-tasks/task_70/file_0.json
> JSON content: [{'id': 0, 'value': 0.012100742504144613}, {'id': 1, 'value': 0.24189955806852526}, {'id': 2, 'value': 0.6917152838781586}, {'id': 3, 'value': 0.5472902176020309}, {'id': 4, 'value': 0.4935773497386685}, {'id': 5, 'value': 0.8534667542973725}, {'id': 6, 'value': 0.11561159547475086}, {'id': 7, 'value': 0.04458665277203899}, {'id': 8, 'value': 0.7982278730716759}, {'id': 9, 'value': 0.0345809384420549}]
> Index: 1
> ...
> ```
>
> ### **(2) Prevent the Failure Within Each Step:**
> We introduced self-debugging mechanism in each subtask that triggers when all candidate API invocation fail. This can prevent the occurence of API failures within intermediate steps, thereby mitigating the propagation of such API errors. Among the erroneous subtask in CoAE, we observe averagely 31.9% erroneous subtask are fixed by Self-debugging mechanism. Among these cases, 58.1% eventually derived successfully executable final solution, 31.3% passed all the test cases.
>
> Through the above two design, ExploraCoder mitigates the propagation of errors in multi-unseen-API task.
> Moreover, due to the reasoning attempts performed at each step in our ExploraCoder, even if certain intermediate steps fail—potentially leading to some missing functionalities in the final solution—the resulting code is still highly executable and impllements most of the required functionality. Such **partially correct code solution** are still of significant valua for developers. In this regard, the usability even in failure cases clearly differentiates our ExploraCoder with end-to-end generation baselines that often produce code with pervasive API misuse patterns.

---

### Note · Authors · 2024-12-07

I have read and agree with the venue's withdrawal policy on behalf of myself and my co-authors.